# BRIDGE: Bootstrapping Text to Control Time-Series Generation via Multi-Agent Iterative Optimization and Diffusion Modeling

**Hao Li** [* 1 2]   **Yu-Hao Huang** [* 1 3]   **Chang Xu** [1]   **Viktor Schlegel** [2 4]
**Renhe Jiang** [5]   **Riza Batista-Navarro** [2]   **Goran Nenadic** [2]   **Jiang Bian** [1]

## Abstract

Time-series Generation (TSG) is a prominent research area with broad applications in simulations, data augmentation, and counterfactual analysis. While existing methods have shown promise in unconditional single-domain TSG, real-world applications demand for cross-domain approaches capable of controlled generation tailored to domain-specific constraints and instance-level requirements. In this paper, we argue that text can provide semantic insights, domain information and instance-specific temporal patterns, to guide and improve TSG. We introduce "Text-Controlled TSG", a task focused on generating realistic time series by incorporating textual descriptions. To address data scarcity in this setting, we propose a novel LLM-based Multi-Agent framework that synthesizes diverse, realistic text-to-TS datasets. Furthermore, we introduce BRIDGE, a hybrid text-controlled TSG framework that integrates semantic prototypes with text description for supporting domain-level guidance. This approach achieves state-of-the-art generation fidelity on 11 of 12 datasets, and improves controllability by up to 12% on MSE and 6% MAE compared to no text input generation, highlighting its potential for generating tailored time-series data. Our code is available at: `Microsoft/TimeCraft`[1].

## 1. Introduction

High-quality Time Series Generation (TSG) is an important task in various domains, including finance (Sezer et al., 2020), healthcare (Li et al., 2023) and environmental science (Hasnain et al., 2022). For example, realistic synthetic medical electrocardiogram (ECG) patterns can be used to train medical residents (Pöhl et al., 2025; Hong & Chun, 2023), while simulating regional electricity usage can be used to stress test the power grid (Westgaard et al., 2021). Although some remarkable works (Huang & Deng, 2023; Bao et al., 2024) have been done for TSG, showing promising results in generating realistic and coherent time series (TS), most of them focus on the basic setting—unconditional single domain generation. However, in real application scenarios, there are specific constraints or requirements for the generated TS to be met, such as specifying domain-specific characteristics, incorporating prior knowledge (Yuan & Qiao, 2024), or satisfying operational constraints (Coletta et al., 2023). For instance, it may be necessary to generate ECG patterns that respect individual patient profiles or capture specific disease conditions (Schlegel et al., 2023). Therefore, the current status in TSG, while demonstrating strong foundational performance, leaves a significant gap for addressing more complex, constrained generation tasks that are crucial for real-world applications.

Some prior work on cross-domain TSG has explored various ways to meet specific generation needs, with most focusing on leveraging domain information to control the generation process. Some approaches rely on explicit domain labels during training (Huang et al., 2024; Kollovieh et al., 2024), treating the task as a conditional generation problem. This allows users to specify the domain during inference. However, this method is limited as it struggles with *unseen* domains and becomes inefficient when the number of domains is large. Other methods incorporate specific information through natural language (Zhou et al., 2024; Liu et al., 2024d), but they operate at the domain level, thus failing to provide detailed fine-grained and instance-specific control, which is essential for more accurate and tailored TSG, highlighting a significant gap in the field.

In this work, we investigate the challenging yet practical research problem of achieving *instance-level* controlled TSG capable of *generalising to unseen domains*. Inspired by the recent success of controlled content generation in images

---

[*]Equal contribution [1]Microsoft Research [2]The University of Manchester, UK [3]Nanjing University, China [4]Imperial College London, Imperial Global Singapore, Singapore [5]The University of Tokyo, Japan. Correspondence to: Chang Xu <chanx@microsoft.com>.

*Proceedings of the 42ⁿᵈ International Conference on Machine Learning*, Vancouver, Canada. PMLR 267, 2025. Copyright 2025 by the author(s).

[1]Work done during Hao Li and Yu-Hao Huang's research internship at Microsoft Research.

(Zheng et al., 2023) and videos (Liu et al., 2024e), where texts are used as a source of control which facilitates capturing complex patterns and semantic relationships, we argue that *using text to provide semantic insights—such as domain information and instance-specific temporal patterns—could enhance and guide TSG*. However, using text for controlled TSG presents two key challenges that need to be addressed in order to fully leverage its potential.

*(i)* **Limited availability of high-quality text-TS pairs**: To train a TSG model that can be controlled by text, we require paired data, where each time series is associated with a detailed text description. However, most available text data only provides high-level domain descriptions, lacking granular, instance-specific information such as trends, fluctuations, or the behavior of individual data points (Liang et al., 2024). The first challenge, therefore, is determining *what specific text information is useful for controlling generation* and *how to obtain such detailed text*. We explored rule-based methods to generate individual-level text descriptions (Harris & Zaki, 2022), but it did not lead to significant performance improvements (as shown in Appendix A.8), suggesting that a more sophisticated approach is needed.

*(ii)* **Bridging discrepancy between text and time-series data for controlled TSG**:

Text and time-series (TS) data exhibit significant differences in both modality and granularity. Text conveys information via a fixed vocabulary of discrete tokens, while time series data is continuous, which leads to inherent mismatches. This disparity may render text too coarse to fully capture domain-specific patterns and characteristics, posing challenges in achieving precise domain-level control due to its incomplete or oversimplified representations. Meanwhile, text can provide detailed, instance-specific descriptions that are crucial for nuanced control, requiring careful and dedicated design to align text with TS features effectively.

**To address the first challenge**, we propose a role-based collaborative multi-agent framework designed to generate high-quality datasets for text controlled TSG. We argue that the process of automatically identifying textual descriptions for TS parallels prompt optimization for large language models (LLMs), where variations in prompt design significantly affect performance (T et al., 2024). Inspired by the success of prior work (Zhou et al., 2023b; Liu et al., 2024c; Guo et al., 2024), our framework consists of three key components: *Text Template Generation*, *Automatic Evaluation*, and *Feedback-driven Refinement*. This process ensures continuous improvement through feedback and synthesis, achieving a holistic optimization of the generated text tailored to the TS at hand. Experimental evaluations reveal the effectiveness of the proposed framework, achieving at least a 15% performance improvement in MAE compare to the unrefined text, while producing outputs that are notably more

comprehensive than those generated through traditional text-generation methods.

**To address the second challenge**, we adopt a hybrid text-enhanced time-series generation strategy. This approach incorporates semantic prototypes (Huang et al., 2025) to extract implicit domain features from TS, complementing text-based conditioning that provides explicit domain information for diffusion model. As a result, the proposed model achieves SOTA performance across multiple datasets on fidelity of the generation results and demonstrates controllability in both in-domain and out-of-domain settings.

To summarize, this paper presents the following novel contributions: **First**, we introduce a multi-agent framework for creating a text-controlled TSG dataset. Our numerical experiments show that textual descriptions provide valuable information for time-series models. **Second**, using this dataset, we analyze the impact of different types of time-series descriptions, advancing the understanding of how LLMs can assist in time-series prediction and generation. **Third**, we propose BRIDGE, a novel framework for text controlled TSG through diffusion models. Our approach excels in generating highly controllable time-series, outperforming baseline models across 11 out of 12 datasets, and highlights its potential for tackling complex, real-world tasks, with promising applications in healthcare, finance, and beyond.

## 2. Related work

**Using Text for Time Series Modeling:** Text-based approaches have shown promise in enhancing various time series tasks, including forecasting (Jin et al., 2023; Gruver et al., 2023; Chen et al., 2023; Xue & Salim, 2024; Zhang et al., 2024b), classification (Xie et al., 2023; Lopez-Lira & Tang, 2023), and event prediction (Gunjal & Durrett, 2023; Shi et al., 2024). While these studies primarily focus on leveraging text to guide or interpret existing time series, text-to-time series generation remains underexplored. GenG (Zhou et al., 2024) initiates this direction with a two-stage pipeline, but is limited to specific domains and lacks instance-level control. Time-MMD (Liu et al., 2024a) presents a large-scale paired dataset for multi-domain text–time series forecasting. However, its step-level annotations focus on local transitions, limiting its applicability to modeling or evaluating alignment over extended temporal sequences.

**Conditional Time Series Generation:** Diffusion-based models have emerged as a powerful paradigm for conditional time series generation. A number of recent works employ denoising score matching and continuous-time diffusion to enable flexible, probabilistic generation (Tashiro et al., 2021; Shen & Kwok, 2023; Narasimhan et al., 2024; Huang et al., 2024; Kollovieh et al., 2024; Shen et al., 2024;

Fan et al., 2024; Deng et al., 2025; Hou et al., 2024) These methods explore a range of network backbones, including score-based transformers (Yuan & Qiao, 2024), structured state space models (Alcaraz & Strodthoff, 2023), and constrained denoising objectives (Coletta et al., 2023). Some also incorporate seasonality and trend decomposition as explicit inductive biases (Yuan & Qiao, 2024). Despite their expressiveness, most models operate within single-domain scenarios and rely on fixed conditioning formats. One of the works with cross-domain focus is TimeDP (Huang et al., 2025), which introduces the use of prototypes to build 'soft prompts" for guiding cross-domain generation. However, its coarse-grained control makes it challenging to achieve personalized text-controlled TSG.

## 3. Problem Formulation

Time series often span multiple domains, each with unique temporal dynamics. Let $D = \{D_1, D_2, \ldots, D_k\}$ denote a set of domains, where each domain $D_i$ is associated with a collection of time series $X_i = \{\mathbf{x}_t \mid t = 1, 2, \ldots, T\}$, and $\mathbf{x}_t \in \mathbb{R}^d$ represents a $d$-dimensional vector at timestamp $t$. Our aim is to generate realistic time series while capturing domain-specific patterns and adhering to constraints specified by textual descriptions. Formally, the goal is to learn a generative function: $f_\theta : (D, l, z) \rightarrow \hat{\mathbf{x}}$, where $l$ represents a textual prompt describing characteristics such as trends or periodicity, and $z$ is a latent variable sampled from a prior distribution. The output $\hat{\mathbf{x}}$ is a time series that aligns with both domain-specific patterns and textual conditions.

Adhering to the channel-independent setting (Nie et al., 2023) that is widely accepted by recent researches, we formulate the problem studied in this paper in a uni-variate time series generation manner to handle the heterogeneity of time series in terms of dimension (Woo et al., 2024).

## 4. Methodology

We propose a unified framework for text-controlled time series generation, which consists of two main stages: *text-to-time series data preparation* and *text-to-time series data generation*. The overall architecture is illustrated in Figure 1, where both stages are tightly coupled to facilitate high-quality, controlled time series synthesis.

### 4.1. Text-to-Time Series Data Preparation

To address the scarcity of high-quality text-to-time series paired datasets, we investigate how to generate effective textual descriptions of time series data to create high-quality TS-text-paired datasets. Initially, we explored some straightforward methods, such as rule-based approaches that relied on simple trend-related terms (e.g., "increasing", "decreasing") and degree modifiers (e.g., "significant", "slight").

Additionally, we tried to leverage GPT-4o and incorporated Seasonal-Trend Decomposition using Loess (STL) (Cleveland et al., 1990) to preprocess time-series data by decomposing it into trend, seasonal, and residual components, making the TS data easier for the model to interpret. However, the resulting texts were overly simplistic and failed to capture the complex patterns and domain-specific nuances inherent in time-series data (results in Appendix A.8). Therefore, we further explore leveraging more diverse and enriched sources to generate text capable of effectively assisting time-series tasks.

To this end, We propose a multi-agent system to automatically generate and iteratively refine TS textual descriptions. As shown in Figure 1, the proposed framework comprises three key components: *Step 1: Text Template Generation*, which focuses on collecting and extracting text templates to construct initial textual descriptions; *Step 2: Automated Evaluation*, designed to assess the effectiveness of the descriptions in supporting downstream tasks; and *Step 3:Feedback-Driven Refinement*, which improves the textual descriptions based on evaluation metrics. Steps 2 and 3 will alternate and iterate until the agent system determines the output is sufficiently refined or a predefined iteration limit is reached. Throughout the iterations, the agents refine a set of general-purpose text templates, which are designed to be dataset-agnostic and easily adaptable to new domains and datasets. Subsequently, these refined templates are utilized to generate textual descriptions for TSG tasks.

**Step 1 Text Template Collection:** As noted in previous work (Merrill et al., 2024), generating fine-grained text descriptions remains a challenging task due to the limited availability of extensive data resources, while also avoiding the potential information leakage that could occur when interacting with external sources while generating text descriptions for a single data. To address this limitation, we adopt a template-based approach that standardizes the narrative of key TS information. Starting with a variety of initial queries, we first collect articles, news, and reports that describe TS data. Inspired by ReAct (Yao et al., 2023), we propose a single-agent framework, which prompts LLMs to generate dynamic reasoning traces for collecting candidates and actions to interact with external environments (e.g., Google, Wikipedia) in an interleaved manner (Madaan et al., 2023) (Framework pipeline can be find in Appendix A.3). The agent decomposes the query into sub-questions, using external tools to answer each sub-question iteratively until all are addressed. Afterwards, another LLM extracts general TS templates from the collected documents, thus curating a set of 50 general-purpose templates. To ensure broad applicability across domains, dataset-specific details are carefully excluded through a combination of prompting techniques and human verification. During the dataset construction phase, an LLM is employed to fill these templates with

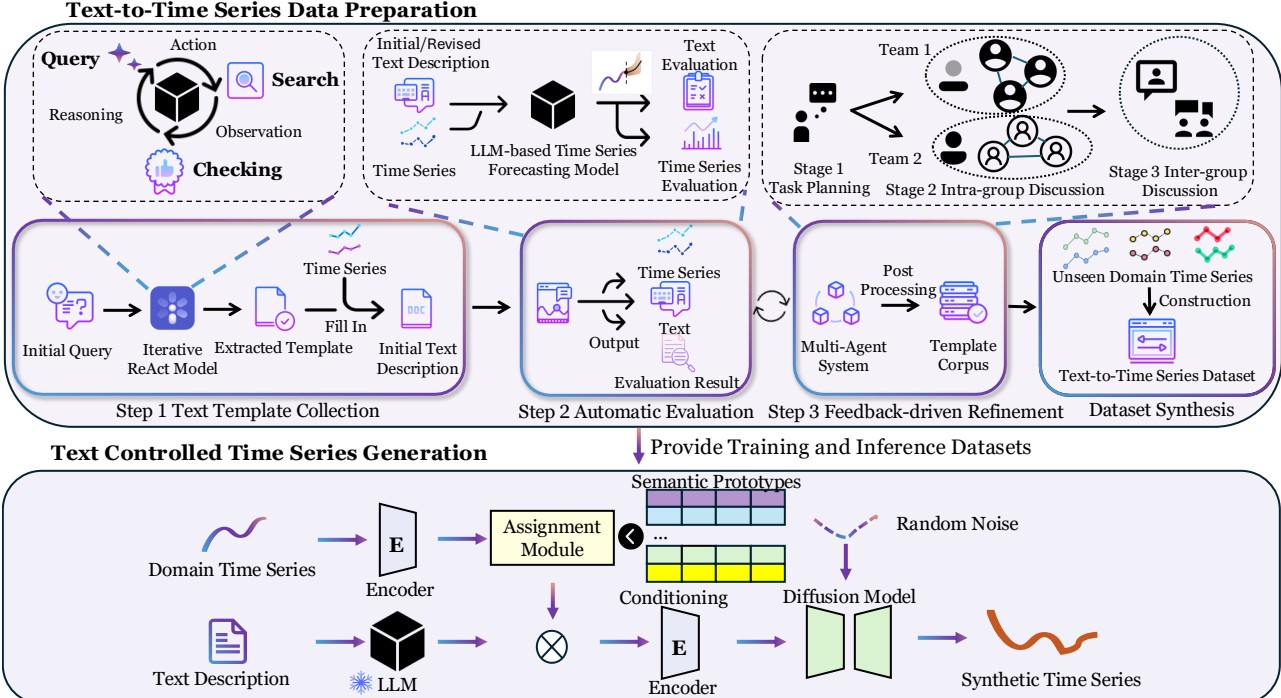

Figure 1: Overview of the proposed BRIDGE framework, which consists of two main stages: *text-to-time series data preparation* and *text-to-time series data generation*. The former consists of three steps: *(1)* collecting initial text templates from online sources, *(2)* performing automatic evaluation to assess the quality of generated descriptions, and *(3)* employing a feedback-driven refinement process via a multi-agent system to iteratively improve the textual outputs. The latter, i.e., the diffusion-based generation model, we adopt a hybrid strategy where *(i)* a textual description is used as input. Meanwhile, *(ii)* we utilize the target domain's time series and extract its corresponding domain prototypes and their weights using Prototype Assignment Module. These prototypes and weights are also used as inputs and are fused with the textual description. This fusion serves as the conditioning input for the diffusion model, which is applied to perform text-controlled TSG.

domain-specific and time-series-related details, resulting in initial textual descriptions tailored to the target domain. The detailed example can be viewed in Appendix A.6.

**Step 2 Automatic Evaluation:** Our goal is to equip the system with the ability to evaluate generated text and provide iterative feedback, aligning with the broader objective of leveraging text to guide TSG. This is achieved by incorporating a ts task conditioned on accompanying text, based on the premise that higher-quality text improves performance. While a straightforward approach might involve pre-training or fine-tuning existing models specifically for TSG, this would demand substantial computational resources. Such an approach would be highly resource-intensive, as it would require training or fine-tuning at every iteration.

Because recent advancements in prompting strategies have demonstrated the potential of LLMs for zero-shot Time Series Forecasting (TSF) (Gruver et al., 2023), we use LST-Prompt (Liu et al., 2024b) and LLMTime (Gruver et al., 2023) as our evaluation backbone for a more compute-efficient alternative to model training. Specifically, we

prompt off-the-shelf LLMs with chain-of-thought (CoT) reasoning (Wei et al., 2022), which enables the integration of text as an additional input modality (Liu et al., 2024b). For the detailed automatic evaluation pipeline, initial evaluation criteria and definitions, refer to Appendix A.4 and A.5. Additionally, we conducted evaluations on the impact of different text types on TSG—results shown in Appendix J demonstrate that generated texts that improve LSTPrompt performance in TSF, also enhances TSG performance.

**Step 3 Feedback-driven Refinement:** Based on the initial text templates and the automatic evaluation system, we further propose a multi-agent collaboration system that simulates the iterative refinement process of a team of human prompt engineers, leveraging the demonstrated capability of LLMs to improve their own outputs (Zhang et al., 2024a). As illustrated in Figure 1, the system operates through three stages: **Stage 1 Task Planning** (assigning tasks and monitoring progress), a manager agent orchestrates the workflow, laying the groundwork for all subsequent stages. This agent assigns specific tasks to independent teams, monitors

progress, and ensures alignment across iterations. **Stage 2 Intra-group Discussion** (independent teams iteratively refining outputs), two independent teams work iteratively to refine the text. Each team includes agents with specific roles: a planner who coordinates tasks, a scientist who analyzes and identifies improvements, an engineer who implements changes, and an observer who critiques the outputs. The teams refine their work through cycles until a satisfactory output is produced. **Stage 3 Inter-group Discussion** (collaborative consensus building), the two teams' leaders, moderated by the manager, engage in a structured dialogue to compare and integrate their results. This collaborative process continues until a consensus is reached, producing a unified output. Once all initial templates are refined, the system extracts them into a new set of templates in **Stage 4 Post Processing**. These final templates are filtered to remove duplicates and ensure they do not contain any dataset-specific or time-series-specific information, resulting in a generalized template library. This library is fixed and remains unaltered during subsequent dataset construction, ensuring consistency and reusability. Further details about the system structure, processing pipeline, and output examples can be found in Appendices A.1, A.2 and A.7.

**Dataset Synthesis:** Excluding domain- or dataset-specific information during template construction ensures their generalizability to unseen domains. To demonstrate this, templates generated with the methodology described in the previous subsection used only two datasets and are applied to 12 additional, entirely disjoint datasets in our TSG experiments (seen Section 5.2). During the dataset construction process, information specific to individual time series is extracted, while domain-specific details, which are pre-defined in the dataset, remain consistent across all instances. Specifically, a standalone LLM, separate from the multi-agent framework, is responsible for extracting statistical information from the time series and populating the templates accordingly. This process is entirely offline, with no reliance on external networks or the LLM's pre-trained knowledge. Once populated, the dataset remain static and unchanged throughout the TSG phase.

## 4.2. Text Controlled Time Series Generation

Although text provides explicit details on trends, statistical properties, and domain-specific information, its discrete nature contrasts with the continuous structure of time-series data, posing challenges for using text as a control mechanism. To address this, we present our framework for text-controlled TSG. As shown in Figure 1, the framework employs a hybrid prompt that integrates *semantic prototypes* which supplement coarse-grained domain descriptions provided by text, along with textual descriptions. This design captures shared patterns across domains and improves the model's generalization ability in unseen domains.

We selected diffusion models (Ho et al., 2020; Lin et al., 2024) as the backbone of our framework due to their proven ability to generate diverse, high-quality data while effectively capturing complex data distributions. Recent successes of diffusion models in time-series forecasting (Rasul et al., 2021) and generation (Qian et al., 2024) further underscore their suitability for this task. The specific design of our architecture is detailed in Appendix B.2.

### 4.2.1. DOMAIN-SPECIFIC PROTOTYPE MATCHING

Relying solely on text descriptions often leads to insufficient representation of domain-specific patterns due to their coarse-grained and abstract nature. To address this, we draw inspiration from TimeDP (Huang et al., 2025), which introduces an automated approach to extracting TS domain features. We propose a hybrid representation strategy that complements domain information using semantic prototypes. Semantic prototypes serve as "bases", representing elementary features of time series such as trends and seasonality. These bases act as a shared "dictionary" across different domains, with each prototype vector serving as a "word" that encodes specific semantic features of TS data.

Specifically, we introduce a set of vectors as time series prototypes $\mathcal{P} \in \mathbb{R}^{N_p \times d}$ for representing cross-domain time series common knowledge, where each prototype vector $\boldsymbol{p} \in \mathbb{R}^{1 \times d}$ serves as the representation of a time series basis and $N_p$ is the number of prototypes. Each time series sample corresponds to a distinct allocation of these bases. The mapping from time-series samples to their respective allocations highlights the importance of prototypes for individual instances and distinguishes among domains. We propose *Prototype Assignment Module* to extract domain-specific prototype weights $\boldsymbol{m}$, with domain characteristics expressed through the selection and weighting of these bases. Then, the extracted prototypes $\mathcal{P}$ and their corresponding weights $\boldsymbol{m}$, denoted as $(\mathcal{P}, \boldsymbol{m})$, are leveraged as supplementary information to enhance the text domain representation, enriching its semantic understanding and cross-domain adaptability. During inference, samples from the target domain are used to extract prototypes and compute weights.

### 4.2.2. MODEL TRAINING

We integrate the semantic prototypes $\boldsymbol{p}$ with weights $\boldsymbol{m}$ and the embeddings of textual descriptions $l$ to construct a hybrid prompt. This prompt is then fed into the cross-attention layers of diffusion-based generation model, enhancing the expressiveness of textual inputs. This enables precise and effective controlled generation. With the conditional denoising mechanism, the denoising objective using $\epsilon$-parameterization (Ho et al., 2020) can be written and simplified as:

$$L = \mathbb{E}[\|\boldsymbol{\epsilon} - \hat{\boldsymbol{\epsilon}}\|^2]$$
$$= \mathbb{E}_{\boldsymbol{x}_0 \in D^T, \boldsymbol{\epsilon} \sim \mathcal{N}(\mathbf{0}, \mathbf{I}), n}[\|\boldsymbol{\epsilon} - \boldsymbol{\epsilon}_{\theta, \boldsymbol{P}}(x_n, n, \boldsymbol{m}, \boldsymbol{l})\|^2] \quad (1)$$

where $n$ denotes the denoising step. Please refer to Section 3 for symbol definitions. For implementation details, see Appendix B.1.

## 5. Experiment Setup and Result Analysis

Our experiments are designed to investigate the key claims of this paper: The feasibility of using text for TSG with BRIDGE, its generalisability to unseen domains (Section 5.2), and its capability to provide instance-level control in both in-domain and out-of-domain scenarios (Section 5.3). We also conduct a comprehensive analysis of textual description types and agent strategies (Section 5.4 and 5.5), as well as investigate the impact of BRIDGE's parameter choices and configuration settings (Section 5.6).

### 5.1. Experimental Setup

#### 5.1.1. BASELINES

We select SOTA methods for both TS generation and forecasting tasks as baselines. For generation, we explore the performance of BRIDGE by comparing with conditional (TimeVQVAE, Lee et al. 2023) and unconditional approaches (TimegGAN, Yoon et al. 2019; GT-GAN, Jeon et al. 2022; TimeVAE, Desai et al. 2021). For forecasting, our goal is to establish the realism of synthetic data. Here, we compare the performance of Time-LLM (Jin et al., 2023), LLM4TS (Chang et al., 2023) and TEMPO (Cao et al., 2024), GPT4TS (Zhou et al., 2023a). Detailed descriptions can be found in Appendix C.2. More details about Experiment Setup and Implementation can be found at Appendix D and Appendix E.

#### 5.1.2. DATASETS

We employ AirPassenger and Sunspots as benchmark datasets for multi-agent system assessing the text types impact. We evaluate the effectiveness of BRIDGE on 12 in-domain datasets including Electricity, Solar, Wind, Traffic, Taxi, Pedestrian, Air, Temperature, Rain, NN5, Fred-MD, Exchange. These datasets have been widely used as benchmark datasets for TSG tasks and obtain from GluonTS (Alexandrov et al., 2020) and Monash Time Series Forecasting Repository (Godahewa et al., 2021). For the evaluation of unknown domains, we selected the stock and web dataset. Finally, We use ILI and M4 (Makridakis et al., 2018) as additional datasets for Time Series Forecasting task.(Details shown in Appendix F.2.)

#### 5.1.3. EVALUATION METRICS

**Assessing Fidelity of TSG:** To evaluate the impact of text types, we use the Mean Absolute Error (MAE). For TC-TSG, we measure the Marginal Distribution Discrepancy (MDD) and Kullback-Leibler (K-L) divergence to quantify the realism of the synthesized data, with further details provided in Appendix G.

**Assessing Controllability of TSG:** To evaluate controllability in text-to-time series generation (TSG), we adopt the **Joint Fréchet Time Series Distance (J-FTSD)** (Narasimhan et al., 2024), which captures both local shape and global distributional similarity between the generated and reference sequences. This metric offers a more faithful measure of alignment than traditional point-wise metrics such as MSE or MAE, especially in cases where semantic similarity does not require exact numerical overlap. However, quantitative metrics alone may overlook perceptual or contextual alignment. To complement this, we conduct a **Human Evaluation**, where outputs are ranked based on how well they reflect the intended text descriptions. We consider two settings: in HE-Rank, annotators evaluate only the outputs generated by different model settings along with the ground truth time series; in HE-Mixed, these candidates are presented alongside three randomly selected distractor time series from the test set. This design enables both focused and distractor-aware evaluation, capturing subtle differences (e.g., constant shifts) while assessing trend or pattern alignment. Results are reported as average ranks across multiple evaluations. See Appendix H for details.

**Time Series Forecasting:** Following previous work (Wu et al., 2023), we measure the MSE and MAE for long-term forecasting. For short-term forecasting on the M4 benchmark, we adopt Symmetric Mean Absolute Percentage Error (SMAPE), Mean Absolute Scaled Error (MASE), and Overall Weighted Average (OWA) as evaluation metrics (Oreshkin et al., 2020).

### 5.2. Overall Generation Performance on Fidelity

#### 5.2.1. PERFORMANCE ON IN-DOMAIN SETTING

**Objective:** We evaluated the overall fidelity of BRIDGE using 12 in-domain datasets. During inference, the text from the test set served as prompts, where the prototypes are generated by the training set.

As shown in Table 1, BRIDGE consistently outperforms existing baselines across a variety of datasets, demonstrating its robustness and versatility. In terms of MDD, BRIDGE (w/o Text) achieves the best performance on most datasets, ranking second only on the Air dataset. For instance, on the Wind dataset, BRIDGE achieves an MDD of $0.316 \pm 0.031$, significantly outperforming models like TimeVAE ($0.943 \pm 0.008$) and TimeGAN ($1.115 \pm 0.159$).

| | Dataset | BRIDGE | BRIDGE w/o Text | BRIDGE w/o Prototype | TimeVQVAE | TimeGAN | GT-GAN | TimeVAE |
|---|---|---|---|---|---|---|---|---|
| **Marginal Distribution Distance** | Electricity | $0.220 \pm 0.070$ | $0.202 \pm 0.066$ | $0.277 \pm 0.068$ | $1.763 \pm 0.088$ | $2.443 \pm 0.765$ | $2.026 \pm 0.280$ | $3.306 \pm 0.044$ |
| | Solar | $375.531 \pm 0.001$ | $375.908 \pm 10.230$ | $376.111 \pm 5.506$ | $466.174 \pm 0.145$ | $460.810 \pm 14.078$ | $476.196 \pm 17.041$ | $365.906 \pm 6.365$ |
| | Wind | $0.316 \pm 0.031$ | $0.319 \pm 0.046$ | $0.362 \pm 0.050$ | $0.777 \pm 0.028$ | $1.115 \pm 0.159$ | $0.706 \pm 0.106$ | $0.943 \pm 0.008$ |
| | Traffic | $0.254 \pm 0.034$ | $0.261 \pm 0.038$ | $0.316 \pm 0.006$ | $1.170 \pm 0.028$ | $1.733 \pm 0.137$ | $1.311 \pm 0.032$ | $0.984 \pm 0.012$ |
| | Taxi | $0.386 \pm 0.057$ | $0.391 \pm 0.057$ | $0.491 \pm 0.133$ | $0.534 \pm 0.032$ | $1.278 \pm 0.168$ | $1.118 \pm 0.157$ | $0.697 \pm 0.007$ |
| | Pedestrian | $0.621 \pm 0.124$ | $0.624 \pm 0.116$ | $0.800 \pm 0.111$ | $1.225 \pm 0.060$ | $1.574 \pm 0.290$ | $1.559 \pm 0.117$ | $0.777 \pm 0.012$ |
| | Air | $0.447 \pm 0.112$ | $0.410 \pm 0.129$ | $0.508 \pm 0.091$ | $0.338 \pm 0.012$ | $2.089 \pm 0.618$ | $2.828 \pm 0.172$ | $1.369 \pm 0.040$ |
| | Temperature | $0.342 \pm 0.010$ | $0.345 \pm 0.019$ | $0.408 \pm 0.053$ | $0.943 \pm 0.035$ | $1.164 \pm 0.110$ | $1.165 \pm 0.072$ | $2.044 \pm 0.024$ |
| | Rain | $5.340 \pm 0.421$ | $5.597 \pm 0.409$ | $6.678 \pm 2.045$ | $9.243 \pm 0.122$ | $10.937 \pm 4.039$ | $6.473 \pm 1.207$ | $9.134 \pm 0.477$ |
| | NN5 | $0.591 \pm 0.029$ | $0.628 \pm 0.021$ | $0.748 \pm 0.134$ | $1.424 \pm 0.043$ | $2.758 \pm 0.142$ | $2.121 \pm 0.094$ | $2.871 \pm 0.045$ |
| | Fred-MD | $0.258 \pm 0.045$ | $0.271 \pm 0.056$ | $0.359 \pm 0.097$ | $2.932 \pm 0.133$ | $4.028 \pm 0.130$ | $4.026 \pm 0.087$ | $2.902 \pm 0.215$ |
| | Exchange | $0.374 \pm 0.053$ | $0.376 \pm 0.058$ | $0.382 \pm 0.041$ | $0.993 \pm 0.058$ | $1.553 \pm 0.122$ | $1.355 \pm 0.072$ | $1.331 \pm 0.042$ |
| **K-L Divergence** | Electricity | $0.011 \pm 0.010$ | $0.014 \pm 0.013$ | $0.013 \pm 0.005$ | $0.185 \pm 0.018$ | $0.395 \pm 0.121$ | $0.415 \pm 0.040$ | $0.580 \pm 0.005$ |
| | Solar | $0.007 \pm 0.002$ | $0.007 \pm 0.003$ | $0.015 \pm 0.012$ | $0.726 \pm 0.043$ | $0.889 \pm 0.288$ | $0.102 \pm 0.045$ | $0.201 \pm 0.008$ |
| | Wind | $0.067 \pm 0.030$ | $0.061 \pm 0.042$ | $0.069 \pm 0.030$ | $0.493 \pm 0.081$ | $4.528 \pm 1.743$ | $0.511 \pm 0.129$ | $0.553 \pm 0.014$ |
| | Traffic | $0.013 \pm 0.004$ | $0.013 \pm 0.005$ | $0.017 \pm 0.004$ | $0.145 \pm 0.015$ | $2.134 \pm 0.952$ | $1.108 \pm 0.171$ | $0.212 \pm 0.006$ |
| | Taxi | $0.013 \pm 0.009$ | $0.013 \pm 0.010$ | $0.039 \pm 0.031$ | $0.100 \pm 0.014$ | $1.160 \pm 0.651$ | $0.663 \pm 0.127$ | $0.120 \pm 0.005$ |
| | Pedestrian | $0.011 \pm 0.009$ | $0.011 \pm 0.009$ | $0.020 \pm 0.005$ | $0.275 \pm 0.021$ | $0.881 \pm 0.436$ | $0.347 \pm 0.085$ | $0.052 \pm 0.010$ |
| | Air | $0.022 \pm 0.017$ | $0.018 \pm 0.014$ | $0.019 \pm 0.011$ | $0.017 \pm 0.004$ | $0.588 \pm 0.369$ | $0.506 \pm 0.091$ | $0.176 \pm 0.016$ |
| | Temperature | $0.023 \pm 0.013$ | $0.024 \pm 0.010$ | $0.039 \pm 0.033$ | $0.980 \pm 0.190$ | $8.775 \pm 2.511$ | $2.177 \pm 0.323$ | $1.910 \pm 0.076$ |
| | Rain | $0.006 \pm 0.001$ | $0.007 \pm 0.001$ | $0.009 \pm 0.003$ | $0.008 \pm 0.002$ | $0.383 \pm 0.089$ | $0.462 \pm 0.056$ | $0.175 \pm 0.011$ |
| | NN5 | $0.010 \pm 0.008$ | $0.013 \pm 0.008$ | $0.016 \pm 0.007$ | $0.603 \pm 0.107$ | $4.054 \pm 1.592$ | $1.372 \pm 0.180$ | $1.284 \pm 0.058$ |
| | Fred-MD | $0.024 \pm 0.019$ | $0.028 \pm 0.029$ | $0.033 \pm 0.029$ | $0.712 \pm 0.054$ | $5.371 \pm 1.455$ | $3.509 \pm 0.299$ | $0.376 \pm 0.025$ |
| | Exchange | $0.083 \pm 0.056$ | $0.088 \pm 0.065$ | $0.088 \pm 0.077$ | $1.984 \pm 0.836$ | $4.376 \pm 0.664$ | $1.583 \pm 0.932$ | $2.011 \pm 0.433$ |

Table 1: Generation results on various univariate datasets, evaluated using Marginal Distribution Distance (MDD) and K-L divergence (K-L). Lower values indicate better performance. Best results are in red, and second best in blue.

The KL divergence results further underscore BRIDGE's capabilities. BRIDGE achieves the lowest KL divergence on almost all datasets, ranking second only on the Wind and Air datasets. On the Electricity dataset, BRIDGE achieves a KL divergence of $0.011 \pm 0.010$, substantially better than BRIDGE (w/o Text) at $0.014 \pm 0.013$, and significantly outperforming baselines such as TimeVQVAE ($0.185 \pm 0.018$) and TimeGAN ($0.395 \pm 0.121$). Notably, even without text conditioning, BRIDGE often secures the second-best performance, highlighting the strength of its core architecture. For example, on the Pedestrian dataset, BRIDGE (w/o Text) achieves the second-best KL divergence of $0.011 \pm 0.009$, only slightly behind the top-performing model. We also show that BRIDGE can generate realistic synthetic data for forecasting downstream tasks in Appendix I.

| | Methods | MDD | | K-L | |
|---|---|---|---|---|---|
| | | 5-shots | 10-shots | 5-shots | 10-shots |
| **Stock** | TimeVQVAE | 3.502 | 3.514 | 1.487 | 3.785 |
| | TimeGAN | 3.834 | 3.765 | 14.347 | 13.823 |
| | GT-GAN | 3.653 | 3.474 | 10.971 | 8.855 |
| | TimeVAE | 3.738 | 3.338 | 6.048 | 4.479 |
| | BRIDGE | 3.477 | 3.112 | 3.249 | 2.827 |
| **Web** | TimeVQVAE | 9.630 | 10.012 | 1.665 | 1.213 |
| | TimeGAN | 8.304 | 8.122 | 4.343 | 3.236 |
| | GT-GAN | 8.018 | 8.936 | 10.037 | 10.915 |
| | TimeVAE | 10.106 | 10.211 | 3.332 | 1.845 |
| | BRIDGE | 8.085 | 7.995 | 0.905 | 0.876 |

Table 2: Few-shot Performance of Unseen domain. We compare the proposed methods and baseline on 5,10-shots. Best results are highlighted in bold face.

### 5.2.2. PERFORMANCE ON UNSEEN DOMAIN SETTINGS

**Objective:** We evaluated the performance of BRIDGE in unseen domains, training on 12 in-domain datasets and testing on two out-of-domain datasets. In this setting, the few-shot scenario established the prototypes, while the text functioned as the control condition for generation .

Our model demonstrates superior robustness in both MDD and K-L metrics, outperforming the baseline models in most cases (Shown in Table 2). Specifically, it achieves the best results for general MDD and K-L scores at both 5-shot and 10-shot settings. Furthermore, the performance improvement with additional examples suggests that the model effectively leverages learned semantic prototypes to recall more accurate domain and pattern information,

enhancing its generalization capability.

### 5.3. Controlling Generation Performance

#### 5.3.1. PERFORMANCE ON IN-DOMAIN SETTINGS

**Objective:** We evaluated the controllability of BRIDGE on 12 in-domain datasets using J-FTSD (Narasimhan et al., 2024) and human evaluation, while keeping all other settings consistent with Subsection 5.2.1.

As shown in Table 3, the proposed BRIDGE consistently outperforms the two ablated variants in most of datasets, according to both the J-FTSD score and human evaluation metrics (HE and HE+3). The removal of textual input leads

| | Electricity | | | Solar | | | Wind | | | Traffic | | |
| | J-FTSD | HE-Rank | HE-Mixed | J-FTSD | HE-Rank | HE-Mixed | J-FTSD | HE-Rank | HE-Mixed | J-FTSD | HE-Rank | HE-Mixed |
|---|---|---|---|---|---|---|---|---|---|---|---|---|
| BRIDGE | 0.538 | 2.3 | 2.4 | 0.295 | 2.6 | 2.8 | 5.011 | 1.8 | 1.8 | 0.570 | 1.4 | 1.4 |
| *w/o Prototype* | 1.164 | 3.3 | 3.4 | 0.322 | 3.2 | 3.3 | 6.843 | 3.1 | 3.2 | 0.597 | 2.1 | 2.1 |
| *w/o Text* | 1.821 | 3.4 | 3.5 | 0.330 | 3.2 | 3.4 | 6.935 | 4.0 | 4.2 | 0.611 | 4.2 | 4.4 |

| | Taxi | | | Pedestrian | | | Air | | | Temperature | | |
| | J-FTSD | HE-Rank | HE-Mixed | J-FTSD | HE-Rank | HE-Mixed | J-FTSD | HE-Rank | HE-Mixed | J-FTSD | HE-Rank | HE-Mixed |
|---|---|---|---|---|---|---|---|---|---|---|---|---|
| BRIDGE | 0.974 | 1.4 | 1.4 | 0.488 | 2.2 | 2.6 | 0.654 | 2.7 | 2.8 | 3.977 | 2.4 | 2.4 |
| *w/o Prototype* | 1.037 | 3.4 | 3.6 | 0.662 | 2.6 | 3.1 | 0.817 | 3.0 | 3.1 | 4.613 | 2.6 | 3.1 |
| *w/o Text* | 1.312 | 4.2 | 4.4 | 0.550 | 3.3 | 4.0 | 0.677 | 3.3 | 3.4 | 4.708 | 3.6 | 3.9 |

| | Rain | | | NN5 | | | Fred-MD | | | Exchange | | |
| | J-FTSD | HE-Rank | HE-Mixed | J-FTSD | HE-Rank | HE-Mixed | J-FTSD | HE-Rank | HE-Mixed | J-FTSD | HE-Rank | HE-Mixed |
|---|---|---|---|---|---|---|---|---|---|---|---|---|
| BRIDGE | 0.147 | 1.9 | 1.9 | 0.972 | 2.5 | 2.8 | 0.260 | 2.5 | 2.8 | 1.581 | 1.9 | 1.9 |
| *w/o Prototype* | 0.141 | 3.2 | 3.2 | 1.115 | 3.1 | 3.5 | 0.341 | 2.7 | 3.0 | 1.846 | 3.1 | 3.1 |
| *w/o Text* | 0.151 | 3.7 | 3.7 | 1.192 | 3.4 | 3.8 | 0.423 | 3.8 | 4.2 | 1.738 | 3.7 | 3.7 |

Table 3: Text control performance of in-domain settings. Measured by J-FTSD and human evaluation.

to the most severe performance drop, particularly in human assessments—demonstrating that text is essential for producing outputs that align with intended semantics. Without textual guidance, the model tends to generate less coherent and less interpretable sequences. For example, on the Traffic and Pedestrian datasets, HE scores increase by over 3 points without text, indicating poor alignment. The "w/o Prototype" setting shows moderate degradation, suggesting that while prototype representations enhance fine-grained alignment and structure, they are less critical than textual descriptions. Notably, the gap is especially visible on datasets with high temporal variability such as Wind and Temperature. These results confirm that both components—text and prototypes—play complementary roles in generating realistic and controllable time series.

### 5.3.2. PERFORMANCE ON UNSEEN DOMAIN SETTINGS

**Objective:** Same to the in-domain setting, we evaluated the controllability of BRIDGE on 2 unseen domain datasets.

Table 4 highlights the structural contributions of text and prototype components in unseen domains. The full BRIDGE model achieves the lowest J-FTSD and most favorable human evaluation scores across both Stock and Web datasets, indicating strong generalization beyond the training distribution. Removing text input causes the most notable performance drop, especially in J-FTSD and HE+3, confirming that textual guidance is critical for semantic alignment and coherent generation. In comparison, the "w/o Prototype" variant shows only moderate degradation, particularly in HE but not in J-FTSD, suggesting that prototypes help enforce structural consistency but are less essential than text for global control. These results support a complementary division of labor: text governs high-level semantics, while prototypes refine local dynamics.

| | Stock | | | Web | | |
| | J-FTSD | HE-Rank | HE-Mixed | J-FTSD | HE-Rank | HE-Mixed |
|---|---|---|---|---|---|---|
| BRIDGE | 7.483 | 2.8 | 2.8 | 5.529 | 2.7 | 3.2 |
| *w/o Prototype* | 7.687 | 2.8 | 3.5 | 5.752 | 3.1 | 3.5 |
| *w/o Text* | 8.178 | 3.4 | 4.0 | 6.302 | 3.2 | 3.4 |

Table 4: Comparison of text control performance of different settings on unseen domains. Measured by J-FTSD and human evaluation.

### 5.4. Performance Analysis on Controlling Text

**Objective:** We conducted a comprehensive analysis of the influence of text types. Specifically, we performed TSF experiments on two benchmark datasets to assess the impact of different text types.

**Conciseness leads to better performance.** Table 5 shows that concise text inputs outperform overly detailed ones, which can mislead the model. This is particularly evident in the case of w/o instance context", where the MAE improves by 1.6 (compared to Initial text") on the AirPassenger dataset, indicating that generating text that fully aligns with human preferences remains a challenging task. Notably, when it comes to longer sequence length, the context provides more useful information (48.64 vs 59.91 on Sunspots). **Clearly specifying the length of the prediction/generation can make the model's performance more stable.** This can be seen from the performance of "w/o statistics". After providing a clear sequence length and statistical values, the model's performance improves. **Background information helps the model.** Similar to the findings of other works (Jin et al., 2023; Merrill et al., 2024), background information can significantly improve the model's performance. This is likely because retrieving the pre-trained knowledge from the LLMs can offer additional contextual information as support. **Direct pattern descriptions are more effective than detailed trend descriptions.** As mentioned in Appendix A.8, when attempting to decompose

| Text Types | AirPassenger | | Sunspots | |
|---|---|---|---|---|
| | LLMTime | LSTPrompt | LLMTime | LSTPrompt |
| *Rule-based Text* | 52.41 | 20.08 | 63.92 | 51.61 |
| *Initial Text* | 49.36 | 15.12 | 59.88 | 49.71 |
| *Refined Text(Ours)* | 40.94 | 12.39 | 48.64 | 42.37 |
| *w/o Instance Context* | 41.96 | 13.54 | 54.33 | 44.23 |
| *w/o Background* | 44.63 | 14.77 | 56.81 | 46.07 |
| *w/o Statistical Context* | 44.01 | 13.41 | 54.24 | 47.12 |
| *w/o Pattern* | 44.36 | 14.52 | 55.16 | 46.84 |
| *w/o Pattern+Statistic* | 44.30 | 14.27 | 56.89 | 45.65 |
| *Baseline (Liu et al., 2024b)* | 45.75 | 15.00 | 59.91 | 47.59 |

Table 5: Zero-shot time-series forecasting performance (Mean Absolute Error, lower is better) using different textual descriptions across two datasets: **AirPassenger** and **Sunspots**.

| Text Types | AirPassenger | | Sunspots | |
|---|---|---|---|---|
| | LLMTime | LSTPrompt | LLMTime | LSTPrompt |
| **Agent Strategies** | | | | |
| *Multi-Agent Teams* | 40.94 | 12.39 | 48.64 | 42.37 |
| *Single (Micro)* | 44.27 | 14.22 | 56.80 | 45.70 |
| *Single (Macro)* | 42.57 | 13.83 | 54.51 | 45.01 |

Table 6: Zero-shot TS forecasting performance under different multi-agent strategies. MAE is reported for each configuration; lower values indicate better performance.

the TS into seasonal, trend, and residual components, the model's performance did not show significant improvement. After multiple iterations, the most effective method was to provide the overall upward/downward trends and explicitly identify the top $k$ extreme points. In addition, we further investigated the impact of revised text types on generation performance. These results support our findings in the TSF task, such as rule-based contextual information may bring more confusion, as detailed in the Appendix J.

### 5.5. Performance Analysis on Individual Components of Multi-Agent System

**Objective:** We further evaluated different agent strategies on refined text. In the strategy experiments, the Macro approach involves a single team making high-level information adjustments, while the Micro approach emphasizes fine-grained details. The Multiple Teams strategy represents a collaborative setting where two teams work together to accomplish the task.

As shown in Table 6, the multi-agent team strategy consistently achieves lower MAE than both micro- and macro-level single-agent approaches across datasets and text types. This suggests that collaborative generation captures a broader range of relevant patterns, leading to more effective forecasting guidance. Between single-agent strategies, macro-level descriptions outperform micro-level ones, indicating that concise, high-level summaries are easier for models to utilize than overly detailed inputs. Notably, all strategies perform better under the LSTPrompt setting, highlighting its stronger alignment with model input expectations. These findings underscore the importance of structured, semantically aligned textual inputs in enabling robust zero-shot forecasting performance.

### 5.6. Ablation Study

**Objective:** We further conducted ablation experiments to explore the impact of different components in the multi-

agent system, as well as the performance of different settings, varying prototype quantities, and using different language models as the text encoder in BRIDGE.

As shown in Table 1, the BRIDGE outperforms other variants on most datasets, as shown by its superior MDD and K-L divergence scores. Removing text input (BRIDGE w/o Text) leads to higher MDD and K-L divergence in nearly all datasets, highlighting the importance of text for improving generation quality. In the NN5 dataset, MDD increases from 0.591 to 0.628, and in the Exchange dataset, K-L divergence rises from 0.083 to 0.088. The removal of prototypes (BRIDGE w/o Prototypes) causes the most significant performance decline. As shown in Table 10 in Appendix K, the number of prototypes significantly influences performance: more prototypes provide richer reference patterns that enhance generation quality. However, we also observe that increasing the number of prototypes beyond 16 brings only marginal improvement. Therefore, we adopt 16 prototypes as a practical trade-off between performance and computational efficiency. Moreover, we explored the impact of pre-training knowledge from LLMs (Dubey et al., 2024). The results show that the larger models have a slight change in performance, but it is not significant, indicating that the pre-training knowledge has a minor influence on performance (shown in Appendix L). For visualization of semantics on various datasets, please refer to the Appendix N.

## 6. Conclusion

In this work, we explore the potential of using text to guide TSG. We propose a multi-agent system for refining textual descriptions and a text-controlled TSG model. Experiments show that concise text improves text-controlled performance, with our model surpassing baselines, especially in few-shot learning, demonstrating strong generalization. Additionally, results indicate that the designed semantic prototypes effectively complemented domain information. These findings lay the groundwork for advancing human-preferred text-controlled TSG.

## Impact Statement

This paper presents work aimed at advancing the field of Time-series Generation (TSG) by introducing a novel framework, BRIDGE, which incorporates text to guide and improve time series generation. The potential societal consequences of our work are significant, as TSG plays a crucial role in applications ranging from simulations to counterfactual analysis. By enabling controlled generation of time series tailored to domain-specific constraints and instance-level requirements, our approach has the potential to drive innovations in fields such as healthcare, finance, and climate modeling. Ethically, the datasets we use are publicly available research datasets that have already undergone ethical review.

## Acknowledgement

Viktor Schlegel is part of the IN-CYPHER programme and is supported by the National Research Foundation, Prime Minister's Office, Singapore, under its Campus for Research Excellence and Technological Enterprise (CREATE) programme. We are especially grateful to Microsoft Research for providing the computational resources that supported this work. We also thank Research IT at the University of Manchester for access to the Computational Shared Facility, and the Imperial College Research Computing Service[2] for additional computing support.

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

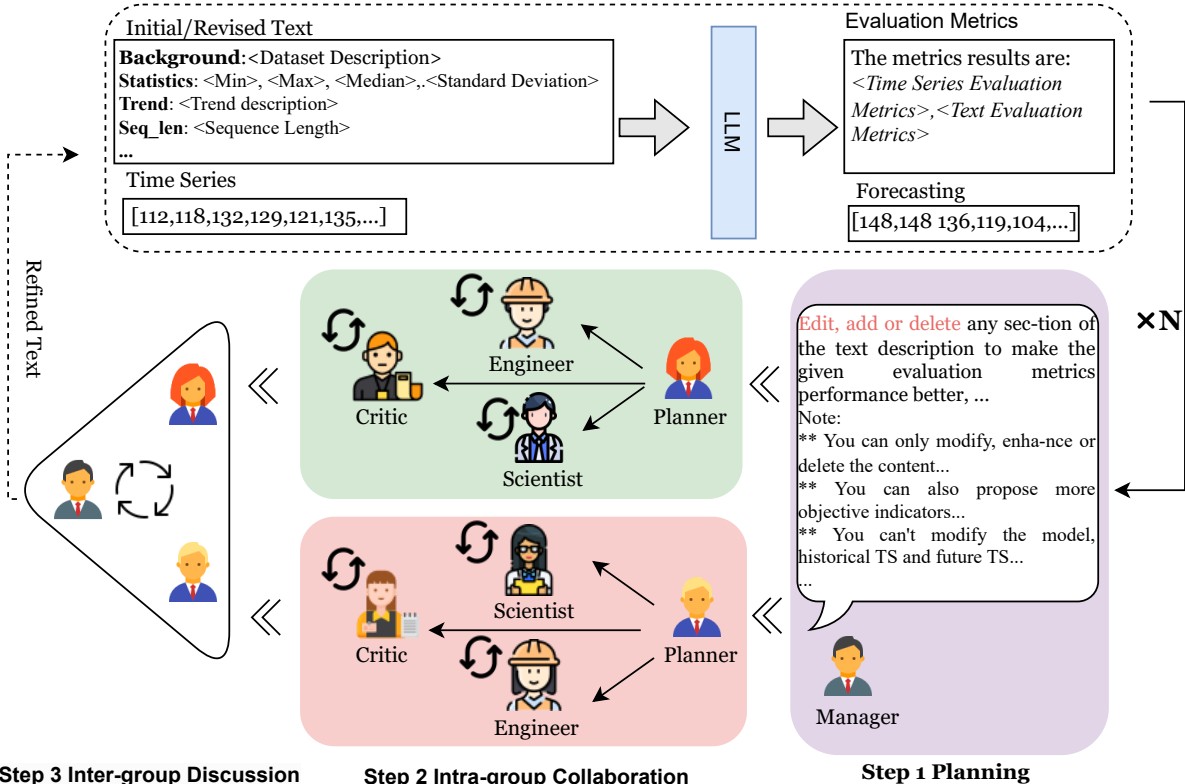

Figure 2: Detail workflow of proposed multi-agent collaborative framework

# A. Text Preparation

## A.1. Multi-Agent Collaboration Framework Details

### A.1.1. FRAMEWORK WORKFLOW

We propose a structured, multi-agent collaboration framework designed to iteratively optimize text generation through systematic refinement. While the system is capable of operating with a single team employing distinct strategies, our experimental results demonstrate that employing two independent teams yields superior outcomes in terms of both quality and diversity of generated outputs. As can be seen from Figure 2, the framework comprises three primary stages:

In Stage 1: Task Planning, a manager agent assumes responsibility for overseeing the workflow. This agent coordinates all subsequent activities by distributing tasks and results from prior iterations to ensure seamless progress and alignment among team members. The manager also defines the objectives for the teams, thereby establishing a structured foundation for collaboration. Stage 2: Intra-group Collaboration constitutes the core of the system, wherein two independent teams of agents work concurrently to refine the given text. Each team is composed of four roles: a planner, a scientist, an engineer, and an observer. The planner serves as the team leader, formulating strategies and supervising operations. The scientist analyzes the input data and formulates detailed optimization plans. The engineer executes these plans, generating improved text outputs. The observer critically evaluates the plans and outputs, raising questions to identify shortcomings and potential improvements. Teams operate in iterative cycles, guided by the observer's critiques. This self-refining loop continues until the observer ceases to raise objections or a predefined maximum number of iterations is reached. Through this iterative process, each team independently produces a refined output. In Stage 3: Inter-group Discussion, the leaders of the two teams engage in a structured dialogue moderated by the manager. This stage facilitates the integration of insights from both teams, encouraging comparative evaluation and collaborative refinement of their outputs. The discussion continues until a consensus is reached, resulting in a unified solution that incorporates the strengths of both teams.

The finalized output is then subjected to Post-Processing. This phase includes a validation step, where the text is evaluated

against a predefined model to ensure its quality and adherence to target metrics. Approved outputs are incorporated into a formal dataset, expanding the training resources available for future tasks. Additionally, any templates developed during the process are added to a general template library, enabling reusability and continuous improvement in subsequent data generation efforts.

A.1.2. STAGES OF COLLABORATION

**Stage 1: Task Planning**
**Role Focus**: Manager

- The manager initiates the workflow and assigns responsibilities to the team leaders.

**Stage 2: Intra-group Collaboration**
**Role Focus**: Planner, Scientist, Engineer, and Observer

- Teams execute their tasks, with internal cycles involving:
    - The scientist proposing plans.
    - The engineer implementing them.
    - The observer providing feedback until quality is satisfactory.

**Stage 3: Inter-group Discussion**
**Role Focus**: Manager and Team Leaders

- Leaders from each team present their refined outputs to the manager.

- Discussions between leaders and the manager lead to a unified, optimized output.

## A.2. Example of Initial and refined text

**Initial Text:** This task focuses on predicting future values of a given time series based on historical data. The historical data shows periodic spikes in values, followed by declines, indicating a strong seasonal pattern. Notable fluctuations are observed at time points when values exceed 400, suggesting external influences. The average value of the historical data is approximately 239.95, with a variance of 8271.86, indicating significant fluctuations around the mean. Future Data Projections indicate that expected values for the time series range between a minimum of 310 and a maximum of 622, with a mean of 0.49 and a standard deviation of 92.75, suggesting increased variability compared to the historical period. The median value is projected to be around 453.34. Anticipated peaks are likely to occur at steps 5, 15, 25, and so on, while predicted dips are expected at steps 8, 18, 28, and similar intervals. Statistical Summary for the future time series reveals the following quartiles: the first quartile (Q1) is **, the median is **, and the third quartile (Q3) is **. Periodicity and Pattern Alignment show consistent high values expected every 10 steps, with auto-correlation anticipated around these intervals, reflecting the cyclical nature observed in the historical data.

**Iteration 1:** This dataset provides monthly totals of US airline passengers from 1949 to 1960. The prediction length is 29 time steps. **Future Data Description:** Future data projections indicate that expected values for the time series range between a **minimum of 310** and a **maximum of 622**, with a **mean of approximately 440.31** and a **standard deviation of about 79.52**, suggesting variability similar to the historical period. The **median value** is projected to be around **419**. The **first quartile (Q1)** is approximately **390.5**, and the **third quartile (Q3)** is approximately **488.5**. Anticipated **peaks** are likely to occur at **steps 5, 15, and 25**, while predicted **dips** are expected at **steps 8, 18, and 28**, reflecting the cyclical patterns observed in the historical data. **Periodicity and pattern alignment** indicate consistent high values expected every **10 steps**, with **auto-correlation** anticipated around these intervals, continuing the cyclical nature observed previously.

**Iteration 2:** Future Data Projections indicate that expected values for the time series range between a minimum of 0.004 and a maximum of 0.98, with a mean of 0.49 and a standard deviation of 0.29, suggesting increased variability compared to the historical period. The median value is projected to be around 0.45. Anticipated peaks are likely to occur at steps 5, 15, 25, and so on, while predicted dips are expected at steps 8, 18, 28, and similar intervals. Statistical Summary for the

future time series reveals the following quartiles: the first quartile (Q1) is 0.21, the median is 0.45, and the third quartile (Q3) is 0.70. Periodicity and Pattern Alignment show consistent high values expected every 10 steps, with auto-correlation anticipated around these intervals, reflecting the cyclical nature observed in the historical data.

**Iteration 3:** The air passengers dataset provides monthly totals of US airline passengers from 1949 to 1960. Data Projections indicate that expected values for the time series range between a minimum of 310 and a maximum of 622, with a mean of 453.34 and a standard deviation of 92.75, suggesting increased variability compared to the historical period. Anticipated peaks are likely to occur at steps 5, 15, 25, and so on, while predicted dips are expected at steps 8, 18, 28, and similar intervals.

**Refined Text:** The air passengers dataset provides monthly totals of US airline passengers from 1949 to 1960. The prediction length is 29 time steps. Data Projections indicate that expected values for the time series range between a minimum of 310 and a maximum of 622, with a mean of 453.34 and a standard deviation of 92.75, suggesting increased variability compared to the historical period. Anticipated peaks are likely to occur at steps 5, 15, 25, and so on, while predicted dips are expected at steps 8, 18, 28, and similar intervals.

### A.3. Pipeline for Collect the text candidate

Figure 3 shows how the single agent framework is proposed how to collect templates. While direct search for relevant content is feasible, it is constrained by a maximum of $K$ titles relevant to the query keyword. To overcome this, we aim to gather relevant candidates based on content similarity. For instance, a simple search for *"time series generation"* might return its definition, but a reasoning-enabled agent can plan what types of articles are more likely to contain relevant content, thereby diversifying the search results.

**Step 1: Single Agent Text Collection**

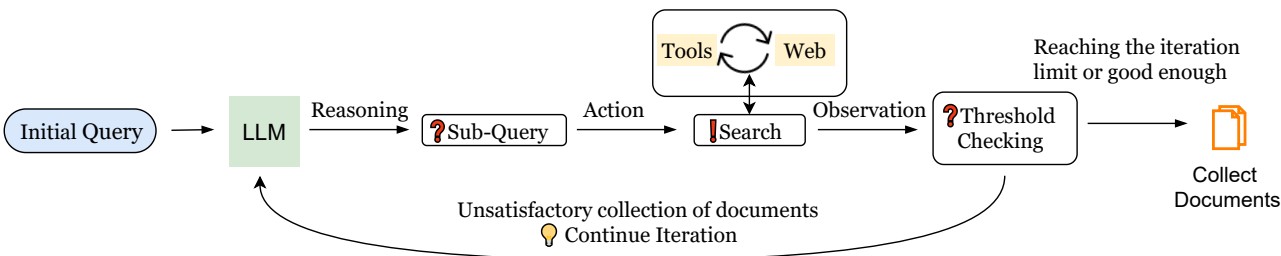

Figure 3: The pipeline of collect the document for extract template. Leverage the ReAct to inspire agent collect human-craft text about time series description. The framework stopped when collect enough document or reach the max iteration limitation.

### A.4. Detail of Automatic Evaluation

The system takes as input a historical TS alongside its textual description, using these as conditions to predict future TS values. This design leverages the intuition that historical TS serves as supplementary contextual information, simplifying the generation process while constraining the model's output space. This framework isolates the text's impact on forecasting accuracy, minimizing the influence of the TS data itself to better assess the quality of the textual input. To validate this approach, we conduct evaluations on two widely studied datasets: the *Air Passenger* and *Sunspots* datasets, which offer diverse temporal patterns and domain characteristics. The key to our refined framework lies in the definition of the evaluation dimension, which directly influences the agent's ability to correct text-time series pair errors and provide high-quality feedback. On one hand, we define evaluation criteria that align with the modal characteristics of both text and TS, allowing the agent to consider both simultaneously in order to correct the data. On the other hand, we also allow the agent to propose more suitable evaluation metrics.

## A.5. Evaluation Dimensions

In this section, we detail the initial evaluation criteria and definitions used to assess the generated text and its impact on TS forecasting. These criteria are designed to align with the modal characteristics of both text and time series, enabling the agent to evaluate and correct the input-output pairs effectively. We consider both text and TS metrics. Specifically, we consider following dimensions for text:

- **Accuracy of trend description:** The description accurately identifies the steady increase in the time series.

- **Mention of seasonality:** The description correctly notes the absence of seasonality in the data.

- **Completeness of information:** The description covers the main aspects of the time series but could mention the exact rate of increase.

- **Clarity of description:** The description is clear and easy to understand.

we consider following for time series: Specifically, we consider Mean Squared Error (MSE) (Hurvich, 1988), Kolmogorov-Smirnov Test (K-S Test) (Berger & Zhou, 2014) and Wasserstein Distance (WD) (Panaretos & Zemel, 2019) for measuring the difference between the generated and target time series, and building a 5-point Likert scale for evaluate the text quality with 5 dimension (i.e. Accuracy of trend description; Mention of seasonality; Reference to external factors; Clarity of description; Completeness of information).

## A.6. Initial Template Exact from Collect Template

The time series templates extracted from the collected corpus typically contain descriptions of key patterns such as trends, seasonalities, and changes over time. For example, a typical template could be structured as:

*Overall, {entity}, {describe_general_trend}. At the beginning, {detail_initial}. As time progressed, {change_description}, culminating in {end_description} by {end_time}*

Additionally, the templates may include other relevant information, such as statistical metrics (e.g., minimum, maximum, standard deviation), dataset information, degree words (e.g., dramatically, slightly) that describe the intensity of changes, and the time series length."

## A.7. Example of Refined Text Template

1.*The dataset_name dataset provides frequency totals of data_description from start_date to end_date. The prediction length is prediction_length time steps. Data statistics indicate that expected values for the time series range between a minimum of min_value and a maximum of max_value, with a mean of mean_value and a standard deviation of std_value, suggesting variability_summary compared to the beginning period. Anticipated peaks are likely to occur at steps peak_steps, while predicted dips are expected at steps dip_steps.*

2.*The dataset_name dataset captures frequency measurements of data_description collected between start_date and end_date. Each series spans prediction_length steps. Statistical indicators show values ranging from min_value to max_value, with an average of mean_value and a variability (standard deviation) of std_value. Notable shifts in the data occur around steps peak_steps (highs) and dip_steps (lows), illustrating variability_summary trends.*

3.*With a focus on data_description, the dataset_name dataset provides frequency records from start_date to end_date. Analysis shows values peaking at max_value and dipping to a minimum of min_value, while maintaining an average of mean_value. The standard deviation of std_value highlights variability_summary. Prediction spans of prediction_length time steps reveal anticipated peaks at peak_steps and dips at dip_steps.*

4.*Designed for generation/forecasting, the dataset_name dataset offers frequency intervals of data_description over a timeline from start_date to end_date. Prediction windows of prediction_length steps enable users to observe patterns such as peak values at peak_steps and dips around dip_steps. Ranges from min_value to max_value suggest considerable variability, with a mean of mean_value and standard deviation of std_value, illustrating variability_summary across the dataset.*

5.*The dataset_name dataset, spanning start_date to end_date, provides frequency observations of data_description. Its prediction horizon is set at prediction_length steps. Statistical analysis reveals that the data ranges from a low of min_value to a high of max_value, with a mean of mean_value and a variability of std_value. Peaks are observed at peak_steps, while dips are noticeable around dip_steps, demonstrating variability_summary throughout the series.*

6.*Spanning from start_date to end_date, the dataset_name dataset includes frequency records of data_description. Predictions are made for horizons of prediction_length time steps. The series ranges between min_value and max_value, averaging mean_value with variability marked by a standard deviation of std_value. Anticipated patterns show higher values near peak_steps and lower ones around dip_steps, reflecting variability_summary over time.*

7.*The dataset_name dataset includes domain time series with frequency frequency, spanning from start_date to end_date. The time series contain total_steps time points, with predictions required for prediction_length steps ahead. The dataset's range of values varies between min_value and max_value, with a mean of mean_value and a standard deviation of std_value. The series shows notable fluctuations, with potential peaks observed at peak_steps and troughs around dip_steps.*

8.*The dataset_name dataset's subdataset includes domain time series data collected at frequency intervals, spanning from start_date to end_date. The total number of time steps is total_steps, and the forecast horizon is prediction_length. Data statistics reveal that the values range from a minimum of min_value to a maximum of max_value, with an average of mean_value and a standard deviation of std_value, indicating variability_summary. Notable peaks are anticipated at time steps peak_steps, while predicted dips are expected at dip_steps.*

9.*The subdataset from the dataset_name dataset contains domain time series recorded at frequency intervals, ranging from start_date to end_date. The series consist of total_steps time points, and the forecast length is prediction_length. The time series data spans a range of values from min_value to max_value, with an average of mean_value and a standard deviation of std_value. The data exhibits variability_summary, with potential peaks at peak_steps and troughs at dip_steps.*

## A.8. Using LLM directly for Time Series Forecasting

We first explored whether rule-based text could work here as it sample and easy to implement. But its results show fluctuations within a certain range, which confuses the model.

Example rule-based template like *"The air passengers dataset provides monthly totals of US airline passengers from 1949 to 1960. The data can be described as small consolidation small consolidation small consolidation small consolidation small breakout rebound small consolidation small rounded_bottom small consolidation small downtrend small consolidation small false_breakout rebound small breakout small downtrend slight breakout small downtrend rebound small breakout moderate downtrend small breakout moderate downtrend small false_breakout small breakout moderate downtrend pullback small breakout small false_breakout small double_bottom small breakout significant downtrend."*

We further leverage LLM to optimize it, aiming to make it more narrative-like. For example

*The input time series illustrates a market with frequent minor fluctuations and distinct pattern formations. It begins with a small consolidation, followed by a false breakout and a rounded bottom. After another phase of consolidation, the market experiences a slight rebound, then consolidates again before another rebound. This is followed by a small breakout and the formation of a double bottom pattern. A rounded bottom and descending triangle emerge, succeeded by a pullback. The market then forms another double bottom and rounded bottom, followed by a period of consolidation. Two more rounded bottoms appear before a small breakout and a subsequent rebound. The series concludes with a final phase of consolidation and the formation of a double bottom.*

*The input time series illustrates a market with frequent minor fluctuations and distinct pattern formations. It begins with a pullback, followed by a small false breakout and a rebound. After a brief period of consolidation, the market experiences another pullback and forms a small rounded bottom. This is followed by another phase of consolidation and a second small rounded bottom, leading to a rebound. The market then forms a small rectangle pattern, rebounds again, and creates a small wedge before pulling back once more. Another small rounded bottom forms, followed by a rebound and two consecutive small false breakouts. The series then undergoes another pullback, experiences a small breakout, forms a small rectangle, and pulls back again. Finally, the market rebounds and concludes with a small breakout.*

Directly employed In-context learning (ICL) to activate LLMs for text generation is also considered. In this setup, the time series first adopts Seasonal-Trend decomposition using Loess (STL) (Cleveland et al., 1990), which is a robust method to decompose time series into long-term trend, seasonal, and residual components. Then, descriptions are generated separately for the initial, intermediate, final, and overall trends. It is important to note that this textual description is based on periodicity rather than time, as the time series is more nuanced. Descriptions segmented by time showed erroneous outputs in experiments, particularly in the form of regular fluctuations within specific intervals. For detailed prompt design consult. Example like:

**Initial Phase (roughly first 30-35 points)**: *The series starts with relatively low values, mostly fluctuating between 4 and 6. There's a slight downward trend initially, followed by a period of relative stability. A few short-lived spikes occur (e.g., around points 7-8), but quickly return to the baseline.*

**Transition Phase (around points 35-40)**: *There's a sudden and significant level shift upwards. Values jump from the 4-6 range to the 8-10 range. This marks the most dramatic change in the entire series.*

**Middle Phase (roughly points 40-70)**: *The series settles into a new, higher range, mostly between 9 and 11. There's increased volatility compared to the initial phase. A slight upward trend is visible within this phase. Several cycles of rises and falls occur, but each cycle tends to peak higher than the last.*

**Late Phase (final 30-35 points)**: *The upward trend becomes more pronounced. Volatility increases further, with larger swings between highs and lows. The series reaches its highest points in this phase, with peaks above 12. Despite the higher*

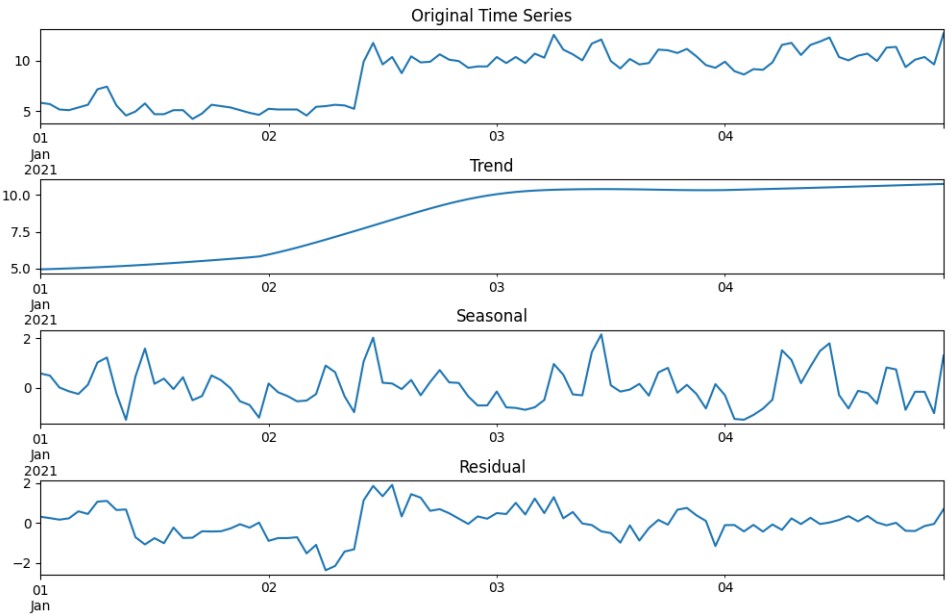

Figure 4: STL Visualization

*peaks, there are still significant drops, creating a saw-tooth pattern.*

**\*\*Overall Trend\*\***: *The series shows a clear overall upward trend from start to finish. This trend is not linear but rather step-like, with a major shift in the middle and then a more gradual increase. The trend is accompanied by increasing volatility over time.*

### A.9. Evaluation

To further ensure the neutrality and safety of the text templates, we conducted the verifications by both automatic and human evaluation: We used GPT-4 to assess whether the templates could be linked to specific TS. Results confirmed that no such linkage existed. Independent human reviewers examined the templates and concluded that they did not reveal any specific details about the TS.

## B. Diffusion-based Time Series Generation

### B.1. Semantic Prototype Assignment and Alignment

**Semantic Prototype Assignment.** Although the same set of prototypes is used across different instances, the degree to which each prototype explains different instances varies. To address this, we assign prototypes to each time series and text description pair, which serves as a condition for the generation model. For each input sequence $x$ (comprising both the time series and text embeddings), a weight vector is generated, the dimension of which corresponds to the number of prototypes. This is achieved via a feature extractor $\phi$. Each element of the vector $\phi(x)_i$ reflects the contribution of each prototype unit $p_i$ in the prototype set $P$, and these weights modify the attention mechanism used during generation. As a result, the model is conditioned on the assigned weighted prototypes. The weights are applied through an attention mask $m$, which operates on the attention weights for prototypes. To ensure sparsity, we discard prototype units that are assigned with negative weights by setting their attention weights to zero. Formally, the prototype assignments are transformed into attention mask $m$ as follows:

$$m = \phi(x_0, t_0) - I_{\phi(x_0,t_0)\leq 0} \cdot \infty \tag{2}$$

where $\phi(x, t) \in \mathbb{R}^{N \times d}$ is the output from the feature extraction layer that processes both time series and text embeddings. $I_{\phi(x,t)\leq 0}$ is an indicator function that zeroes out negative weights, ensuring that only retains non-negative values. In this work, we use randomly orthogonal vectors as prototypes (Huang et al., 2025). In future work, we will consider methods such as fine-tuning.

---

**Algorithm 1** Bridge Training

---

**Require:** Time series-text dataset $\{\mathcal{X}, \mathcal{T}\}$
**Ensure:** Network parameters $\phi$ and $\theta$, prototypes $P$
 1: Initialize prototypes $P$
 2: **repeat**
 3:     Sample $x_0, s$ from $\{\mathcal{X}, \mathcal{T}\}$
 4:     Extract prototype assignments $m = \phi(x_0, P)$
 5:     Randomly set $P$ as unconditional identifier $p_u$
 6:     Randomly sample time step $n \sim \mathcal{U}(1, N)$
 7:     Randomly sample noise $\epsilon \sim \mathcal{N}(0, I)$
 8:     Encode text description: $l = \text{TextEncoder}(s)$
 9:     Corrupt data $x_n = \sqrt{\alpha_n} x_0 + \sqrt{1 - \alpha_n} \epsilon$
10:     Predict step noise with $\tilde{\epsilon} = \tilde{\epsilon}_{\theta, P}(x_n, n, m, l)$
11:     Compute loss and take gradient step
12: **until** maximum training step
13: **Output:** Network parameters $\phi, \theta$ and prototypes $P$

---

**Algorithm 2** Bridge Inference

---

**Require:** Prototypes $P$, time series prompts $x$ from train set or few-shot demonstration set, and text description $s$ from test set
**Ensure:** Generated time series samples $\hat{x}$
 1: Extract prototype prompts $m = \phi(x, P)$
 2: Encode text description $l = \text{TextEncoder}(s)$
 3: Randomly sample noise $\hat{x}_N \sim \mathcal{N}(0, I)$
 4: **for** $n$ from $N$ to $1$ **do**
 5:     Predict step noise with $\tilde{\epsilon}_n = \tilde{\epsilon}_{\theta, P}(\hat{x}_n, n, m, l)$
 6:     Denoise $\hat{x}_{n-1} = \frac{\hat{x}_n - \sqrt{1 - \alpha_n} \tilde{\epsilon}_n}{\sqrt{\alpha_n}}$
 7: **end for**
 8: $\hat{x} = \hat{x}_0$
 9: **Output:** Generated samples $\hat{x}$

---

**Semantic Prototype Alignment.** To condition the denoising diffusion process, we adapt the denoising objective using $c$ as a condition, influencing the model's intermediate layers through cross-attention. This ensures that the generated time series aligns with the specified conditional instruction. To achieve this, we aim to align the condition and semantic prototypes during the training phase. A set number of query embeddings are allocated to both the text and time series as input. These queries interact with each semantic prototype through cross-attention layers (inserted every other transformer block $z$). We initialise the weights of the cross-attention layers randomly and update them during training. Specifically, we apply cross-attention to the feature representations using the following equations:

$$Q = W_Q \cdot cz, \quad K = W_K \cdot \mathbf{P}, \quad V = W_V \cdot \mathbf{P} \tag{3}$$

where

$$z = \text{FF}\left(\text{softmax}\left(\frac{Q(K)^T}{\sqrt{d}} + m\right) \cdot V\right) \tag{4}$$

Here, $z \in \mathbb{R}^{N \times d}$ denotes the output from the attention block. $W_Q, W_K, W_V \in \mathbb{R}^{d \times d}$ are learnable projection matrices applied on the sequence dimension. The attention output $z_{\text{final}}$ is passed to another feedforward network to produce the final output $\hat{\epsilon} = \text{FF}(z_{\text{final}})$.

### B.2. Algorithm for Generation

Algorithm 1 and Algorithm 2 shown the diffusion process of generate new sample with text guide.

## B.3. The form of Input and Output

Input of Diffusion Model including a time series and corresponding text description. The two of them will be processed by the encoder and LLama respectively, and the obtained embedding will be fused through a single-layer MLP as conditional input. The output of the diffusion model is a synthetic time series.

# C. Baseline Model

## C.1. Time Series Generation Model

**TimeVQVAE**(Lee et al., 2023) is a generative model designed for sequential data. It combines the strengths of a variational autoencoder (VAE) with vector quantisation to discretise latent space representations, making it effective for time series data. The model consists of an encoder that compresses the input data into a discrete latent space and a decoder that reconstructs the time series. TimeVQVAE is particularly useful for generating realistic time series samples while maintaining key temporal dependencies. The quantisation step helps in learning discrete representations that can be reused for efficient time series modelling and generation.

**TimeGAN** (Yoon et al., 2019) is a variant of the GAN framework specifically tailored for time series data. It combines both supervised and unsupervised learning approaches, using a generator to create synthetic time series and a discriminator to differentiate between real and generated data. Additionally, it integrates an embedding network to capture temporal dependencies and preserve temporal correlations between generated samples. The model ensures that the generated time series not only closely mimic the statistical properties of the original data but also maintain the correct temporal ordering and dynamics. TimegGAN is particularly useful in applications requiring realistic synthetic data generation, such as forecasting and anomaly detection.

**GT-GAN** (Jeon et al., 2022) introduces a novel architecture for time series generation by incorporating both global and local perspectives. The model features two generators: one focuses on capturing the global trends across the entire time series, while the other focuses on local variations. The two components work together to ensure that the generated time series exhibit realistic patterns on both macro and micro levels. GT-GAN uses a two-stream discriminator that evaluates both the global and local outputs, ensuring high fidelity in the generated data. This model is effective for generating complex time series where both long-term trends and short-term fluctuations are important.

**TimeVAE** (Desai et al., 2021) extends the traditional VAE architecture to model time series data. It uses an encoder to map time series data into a continuous latent space, from which the decoder reconstructs the original time series. The model captures uncertainty and variation in the data through the latent space's probabilistic structure, making it well-suited for applications where capturing latent factors and generating multiple plausible future scenarios is important. TimeVAE can be applied to various tasks, such as anomaly detection, forecasting, and data augmentation, by learning complex temporal dependencies and generating realistic time series that adhere to the original data's statistical properties.

## C.2. Time Series Forecasting Model

**Time-LLM** (Jin et al., 2023) is a powerful TS LLM that outperforms specialized forecasting models, which repurposes LLMs for time series forecasting by reprogramming input data and employing the Prompt-as-Prefix (PaP) technique for enhanced context alignment.

**GPT4TS** (Zhou et al., 2023a) takes advantage of pre-trained language and vision models for general time series analysis. By demonstrating that supervised fine-tuning (SFT) can successfully extend LLM capabilities to time series tasks, GPT4TS bridges the gap between natural language processing models and temporal data analysis. The model's architecture shows the feasibility of applying large pre-trained models to time series, leading to significant performance improvements in various time series applications.

**LLM4TS** (Chang et al., 2023) is an innovative framework that repurposes pre-trained LLMs for time-series forecasting, employing a two-stage fine-tuning strategy and a two-level aggregation method to align with and enhance the model's ability to process multi-scale temporal data, outperforming state-of-the-art models in both fune-tuing and few-shot scenarios.

**TEMPO** (Cao et al., 2024) proposed using prompts to adapt to different time series distributions. It demonstrates superior performance in zero-shot settings across diverse benchmark datasets, showcasing its potential as a foundational model-building framework for capturing dynamic temporal phenomena.

# D. Experiment Setup

The time series length $T$ for generation is set to 168 in a form of non-overlap sequence slices for all the datasets. For forecasting, we assessed performance over four different prediction horizons $H \in \{24, 36, 48, 60\}$ for ILI and $H \in \{6, 48\}$ for M4. The dataset is divided into training, validation, and test sets in a ratio of 8:1:1. All models are tested with the same dataset size, as adding text descriptions to time series sliding is resource-intensive, prompting a reduction in the number of sliding windows.

# E. Implementation Detail

We implemented all the model and conduct all experiments on single NVIDIA Tesla A100 80GB GPUs. For LLM used in proposed model is LLama3-8B (Dubey et al., 2024). For generation task, we keep all model's sequence length is 168 which is the max length of `Pedestrian`, `Rain`, `Temperature` datasets. For evaluation of the synthesis data quality task, we keep the sequence length of 256. The dataset is divided into training, validation, and test sets in a ratio of 8:1:1.

The reported result are all under following training settings. The number of prototypes are set to 16 for all the main evaluations. Models for each sequence length are trained for $50,000$ steps using a batch size of 128 and a learning rate of $5*10^{-5}$ with $1,000$ warm-up steps.

# F. Dataset Analysis

### F.1. Details of Datasets

In this section, we provide a detailed overview of the datasets used for model training in this paper:

**Electricity:** This dataset represents the hourly electricity consumption of 321 clients from 2012 to 2014, measured in kilowatts (kW). It was originally extracted from the UCI repository.

**Solar:** This dataset contains 137 time series representing hourly solar power production in the state of Alabama throughout 2006.

**Wind:** This dataset contains a single extensive daily time series representing wind power production (in megawatts) recorded at 4-second intervals starting from August 1, 2019. It was downloaded from the Australian Energy Market Operator (AEMO) online platform.

**Traffic:** This dataset contains 15 months of daily data (440 daily records) describing the occupancy rate (between 0 and 1) of different car lanes on San Francisco Bay Area freeways over time.

**Taxi:** This dataset contains spatio-temporal traffic time series of New York City taxi rides recorded at 1,214 locations every 30 minutes during January 2015 and January 2016.

**Pedestrian:** This dataset contains hourly pedestrian counts captured from 66 sensors in Melbourne starting from May 2009. The original dataset is regularly updated when new observations become available. The dataset used here contains pedestrian counts up to April 30, 2020.

**Air Quality:** This dataset was used in the KDD Cup 2018 forecasting competition. It contains hourly air quality measurements from 59 stations in two cities: Beijing (35 stations) and London (24 stations) from January 1, 2017, to March 31, 2018. The air quality measurements include various metrics such as PM2.5, PM10, NO2, CO, O3, and SO2. Missing values were imputed using leading zeros or the Last Observation Carried Forward (LOCF) method.

**Temperature:** This dataset contains 32,072 daily time series with temperature observations and rain forecasts gathered by the Australian Bureau of Meteorology from 422 weather stations across Australia between May 2, 2015, and April 26, 2017. Missing values were replaced with zeros, and the mean temperature column was extracted for use.

**Rain:** This dataset focuses on rain data extracted from the same source as the Temperature dataset.

**NN5:** This dataset was used in the NN5 forecasting competition. It contains 111 time series from the banking domain with the goal of predicting daily cash withdrawals from ATMs in the UK. Missing values were replaced by the median across all corresponding days of the week throughout the entire series.

**Fred-MD:** This dataset contains 107 monthly time series reflecting various macroeconomic indicators sourced from the

| Domain | Tasks | Datasets | Dim. | Series Length | Dataset Size | Frequency |
|---|---|---|---|---|---|---|
| Long-Term | ILI | 7 | 24, 36, 48, 60 | (617, 74, 170) | 1 week | Illness |
| Short-term Forecasting | M4-Yearly | 1 | 6 | (23000, 0, 23000) | Yearly | Demographic |
| | M4-Quarterly | 1 | 8 | (24000, 0, 24000) | Quarterly | Finance |
| | M4-Monthly | 1 | 18 | (48000, 0, 48000) | Monthly | Industry |
| | M4-Weekly | 1 | 13 | (359, 0, 359) | Weekly | Macro |
| | M4-Daily | 1 | 14 | (4227, 0, 4227) | Daily | Micro |
| | M4-Hourly | 1 | 48 | (414, 0, 414) | Hourly | Other |

Table 7: Comparison of datasets for long-term and short-term forecasting tasks

Federal Reserve Bank's FRED-MD database. The series have been differenced and log-transformed following established practices in the literature.

**Exchange:** This dataset records daily exchange rates for eight currencies.

**Stock:** This dataset consists of daily stock prices for the symbol GOOG, which is listed on NASDAQ.

**Web:** This dataset was used in the Kaggle Wikipedia Web Traffic forecasting competition. It contains 145,063 daily time series representing the number of hits or web traffic for Wikipedia pages from July 1, 2015, to September 10, 2017. Missing values were replaced with zeros.

### F.2. Dataset Statistics

To test the quality of the synthetic data generated by our proposed model, we conducted tests on two additional datasets. In the experiments, we trained the synthetic data to be the same as the original data and tested it on the real datasets. The statistics of the datasets are in Table 7:

## G. Evaluation Metrics

The calculations of these metrics are as follows:

$$\text{MSE} = \frac{1}{H}\sum_{h=1}^{H}(Y_h - \hat{Y}_h)^2, \qquad \text{MAE} = \frac{1}{H}\sum_{h=1}^{H}|Y_h - \hat{Y}_h|,$$

$$\text{SMAPE} = \frac{200}{H}\sum_{h=1}^{H}\frac{|Y_h - \hat{Y}_h|}{|Y_h| + |\hat{Y}_h|}, \qquad \text{MAPE} = \frac{100}{H}\sum_{h=1}^{H}\frac{|Y_h - \hat{Y}_h|}{|Y_h|},$$

$$\text{MASE} = \frac{1}{H}\sum_{h=1}^{H}\frac{|Y_h - \hat{Y}_h|}{\frac{1}{H-s}\sum_{j=s+1}^{H}|Y_j - Y_{j-s}|}, \qquad \text{OWA} = \frac{1}{2}\left(\frac{\text{SMAPE}}{\text{SMAPE}_{\text{Naïve2}}} + \frac{\text{MASE}}{\text{MASE}_{\text{Naïve2}}}\right),$$

where $s$ is the periodicity of the time series data, $H$ denotes the number of data points (i.e., prediction horizon in our cases), and $Y_h$ and $\hat{Y}_h$ are the $h$-th ground truth and prediction, where $h \in \{1, \ldots, H\}$.

For generation, we consider Marginal Distribution Difference (MDD):

$$\text{MDD}(P, Q) = \sum_{x \in X}|P(x) - Q(x)|$$

where $P$ and $Q$ represent the marginal distributions of the real and synthetic data, and $X$ denotes the set of possible values for the variable being analyzed.

Also Kullback-Leibler divergence (K-L)

$$D_{KL}(P\|Q) = \sum_{x \in X}P(x)\log\left(\frac{P(x)}{Q(x)}\right)$$

| Dataset | Random | | LLM4TS | | TEMPO | | Time-LLM | | GPT4TS | |
|---|---|---|---|---|---|---|---|---|---|---|
| | MSE | MAE | MSE | MAE | MSE | MAE | MSE | MAE | MSE | MAE |
| ILI *Bridge* | | | 1.98 | 0.89 | 1.21 | 1.02 | 2.20 | 1.44 | 2.19 | 1.02 |
| *Real* | 8.12 | 2.14 | 1.86 | 0.86 | 0.96 | 0.82 | 2.00 | 1.20 | 1.90 | 0.90 |
| *KernelSynth* | | | 4.35 | 1.50 | 1.64 | 1.07 | 1.43 | 1.01 | 3.80 | 1.42 |
| | SMAPE | MASE | OWA | SMAPE | MASE | OWA | SMAPE | MASE | OWA | SMAPE | MASE | OWA | SMAPE | MASE | OWA |

| Dataset | Random | | | LLM4TS | | | TEMPO | | | Time-LLM | | | GPT4TS | | |
|---|---|---|---|---|---|---|---|---|---|---|---|---|---|---|---|
| | SMAPE | MASE | OWA | SMAPE | MASE | OWA | SMAPE | MASE | OWA | SMAPE | MASE | OWA | SMAPE | MASE | OWA |
| M4 *Bridge* | | | | 13.10 | 1.99 | 0.93 | 12.25 | 1.73 | 0.87 | 12.85 | 1.90 | 0.96 | 13.10 | 1.99 | 0.93 |
| *Real* | 26.46 | 4.43 | 1.92 | 12.33 | 1.80 | 0.88 | 12.09 | 1.72 | 0.85 | 12.50 | 1.77 | 0.89 | 12.70 | 1.94 | 0.91 |
| *KernelSynth* | | | | 14.39 | 2.09 | 1.02 | 13.94 | 2.00 | 0.99 | 12.594 | 1.735 | 0.918 | 14.56 | 2.08 | 1.02 |

Table 8: Comparison of MSE and MAE for TS forecasting across methods, with results for four forecasting horizons: $H \in \{24, 36, 48, 60\}$ for ILI and $H \in \{6, 48\}$ for M4. Average results are reported.

where $P$ and $Q$ are the two probability distributions being compared, and $X$ represents the set of possible values.

# H. Human Evaluation

In addition to the quantitative metrics, we conducted a human evaluation to assess the quality of the generated time series descriptions.

## H.1. Evaluation Process:

The human evaluation involved a team of trained annotators who were tasked with reviewing the generated time series in relation to the given text descriptions and ranking the candidate. The process was organized into two primary stages:

**Preliminary Setup** Annotators were presented with a single text description accompanied by a set of time series. Each set was evaluated independently, with annotators assessing how well each time series captured the trends, patterns, and anomalies described in the corresponding text. We define two evaluation settings: *(*i) HE-Rank, annotators ranked the time series generated by different model settings together with the ground truth, focusing purely on relevance. *(*ii) HE-Mixed, the same sequences were evaluated alongside three distractor time series randomly sampled from the test set, forming a mixed pool of candidates for relative ranking.

**Evaluation Criteria** To help annotators better understand and standardize the evaluation process, we provide a set of reference dimensions.

- Relevance to given text descriptions: Whether the time series description accurately reflects the patterns, trends, and anomalies observed in the text.

- Semantic Alignment: Whether the time series properly conveys the underlying meaning of the texr, such as identifying upward and downward trends, spikes, and seasonal behaviors.

- Plausibility: Whether the time series makes sense within the domain context (e.g., finance, healthcare), and offers a reasonable interpretation of the data.

- Coherence: Whether the time series is logically consistent within itself, avoiding contradictions and ensuring that the time series aligns with the observed trends or patterns throughout the text.

**Reliability of Evaluation** To ensure consistency and minimize evaluator bias, multiple annotators assessed each time series. The final score for each description was calculated as the average score from all annotators. Annotators were trained on a common set of guidelines to ensure that the evaluation criteria were applied consistently across all descriptions.

# I. Performance of downstream tasks

Table 8 shows the quality of generated data for the purposes of training models for downstream tasks. We generated synthetic data on two additional datasets to assist existing SOTA models in TS forecasting. All models were trained either using only real data or synthetic data and then tested on real test sets. The results indicate that training with only synthetic data can achieve comparable performance to real data across all models, as performance differences between real and synthetic data are less visible than differences in performance between architectures. This suggests that the generated data

is sufficiently realistic, potentially allowing to share synthesised surrogates of otherwise sensitive data. For comparison, we also employed KernelSynth (Ansari et al., 2024) methods. Both methods effectively provided valuable synthetic data (compared to completely random data), but our proposed approach produced data that more closely resembles real data. This underscores its potential for generating meaningful synthetic data across domains.

## J. Ablation Experiment on the Influence of Text Types on Diffusion Models

Table 9 show the performance of different text types. Overall, the inclusion of text descriptions in time series modeling demonstrates consistent benefits across most datasets, with specific nuances depending on the approach and dataset characteristics. Across most datasets, the inclusion of textual descriptions significantly improves alignment metrics, with the Bridge (w/o Background) approach often demonstrating superior performance compared to both Bridge (w/o Pattern+Statistic) and Bridge (w/o Text).

For MDD, Bridge (w/o Background) consistently outperforms or matches other methods in several datasets. For instance, in the "Rain" dataset, Bridge (w/o Background) achieves the lowest score (5.477), outperforming Bridge (w/o Pattern+Statistic) (5.499) and Bridge (w/o Text) (6.002). Similarly, in the "Pedestrian" dataset, Bridge (w/o Background) achieves the best score (0.560), indicating its ability to capture intricate patterns better than the other methods. While Bridge (w/o Pattern+Statistic) occasionally achieves slightly better results (e.g., in the "Electricity" dataset with 0.110 compared to 0.139 for Bridge (w/o Background)), Bridge (w/o Background) demonstrates more consistent performance across diverse domains.

The K-L Divergence results further highlight the strengths of Bridge (w/o Background). In the "Wind" dataset, for example, Bridge (w/o Background) achieves the lowest divergence (0.048), outperforming both Bridge (w/o Pattern+Statistic) (0.052) and Bridge (w/o Text) (0.056). Similar trends are observed in the "NN5" and "Temperature" datasets, where Bridge (w/o Background) consistently produces the lowest K-L scores, indicating its capability to effectively encode temporal patterns with textual assistance. Notably, in datasets like "Air", Bridge (w/o Background) achieves significant improvements compared to Bridge (w/o Text), further emphasizing its advantage in handling complex and noisy time series data.

## K. The Impact of Prototype

The results in Table 10 demonstrate the influence of prototype quantity on marginal distribution distance and K-L divergence across multiple datasets. Increasing the number of prototypes generally leads to improved performance, as observed in the Electricity dataset, where the marginal distribution distance decreases from 0.615 (4 prototypes) to 0.135 (16 prototypes). Similar trends are evident in datasets such as Traffic (1.211 to 0.315) and NN5 (1.550 to 0.613).

For K-L divergence, a higher number of prototypes often results in lower divergence values, indicating better alignment with the target distribution. For example, in the Taxi dataset, K-L divergence drops from 0.154 (4 prototypes) to 0.003 (16 prototypes). However, certain datasets, such as Wind and Exchange, exhibit less consistent trends, suggesting potential variations in data characteristics affecting prototype effectiveness.

## L. The Impact of LLms on the diffusion model performance

The Table 11 compares the performance of Llama and GPT2 as encoders in our diffusion model across various time series domains. Both models show similar performance in most domains, with slight differences in specific cases. For example, Llama performs slightly better in the "Electricity" (0.139 vs 0.174) and "NN5" (0.570 vs 0.887) domains, suggesting a better ability to capture fluctuations in these time series. In contrast, GPT2 outperforms Llama in "Pedestrian" (0.578 vs 0.483) and "Fred-MD" (0.226 vs 0.225), indicating its strength in encoding gradual trends. Overall, both models show strong performance across multiple domains, with only minor variations. These results highlight that while Llama and GPT2 differ slightly in their handling of specific time series patterns, both are effective encoders for our diffusion model, capable of capturing both domain-specific and general temporal features.

## M. Data Augmentation Results

For long-term forecasting (Table 12), we find that the LLM4TS trained via the synthetic data produces relatively low MSE and MAE values, such as ILI-24 Synthesis with an MSE of 1.84 and an MAE of 0.85, which are competitive with the performance on real-world datasets. In fact, for length like 24 and 36, LLM4TS consistently performs well, showing

| | Dataset | w/o Pattern+Statistic | w/o Background | w/o Text | Rule-based |
|---|---|---|---|---|---|
| Marginal Distribution Distance | Electricity | 0.110 | 0.139 | 0.135 | 0.256 |
| | Solar | 375.531 | 375.530 | 375.531 | 377.232 |
| | Wind | 0.344 | 0.316 | 0.304 | 0.422 |
| | Traffic | 0.324 | 0.309 | 0.315 | 1.178 |
| | Taxi | 0.328 | 0.325 | 0.338 | 0.641 |
| | Pedestrian | 0.584 | 0.560 | 0.576 | 1.277 |
| | Air | 0.472 | 0.440 | 0.418 | 0.665 |
| | Temperature | 0.332 | 0.331 | 0.356 | 0.572 |
| | Rain | 5.499 | 5.477 | 6.002 | 9.533 |
| | NN5 | 0.591 | 0.570 | 0.613 | 1.377 |
| | Fred-MD | 0.239 | 0.226 | 0.228 | 0.460 |
| | Exchange | 0.315 | 0.316 | 0.376 | 0.430 |
| K-L Divergence | Electricity | 0.002 | 0.003 | 0.001 | 0.008 |
| | Solar | 0.008 | 0.006 | 0.005 | 0.033 |
| | Wind | 0.052 | 0.048 | 0.056 | 0.144 |
| | Traffic | 0.013 | 0.012 | 0.016 | 0.027 |
| | Taxi | 0.020 | 0.014 | 0.020 | 0.093 |
| | Pedestrian | 0.006 | 0.009 | 0.009 | 0.079 |
| | Air | 0.006 | 0.005 | 0.021 | 0.042 |
| | Temperature | 0.023 | 0.016 | 0.020 | 0.904 |
| | Rain | 0.006 | 0.006 | 0.008 | 0.016 |
| | NN5 | 0.008 | 0.004 | 0.004 | 0.101 |
| | Fred-MD | 0.005 | 0.004 | 0.023 | 0.108 |
| | Exchange | 0.062 | 0.057 | 0.067 | 0.350 |

Table 9: The performance of different text type. Marginal distribution distance scores (MDD) and K-L divergence (K-L) are reported.

| | Prototypes | 4 | 8 | 16 | 32 | 64 | 4 | 8 | 16 | 32 | 64 |
|---|---|---|---|---|---|---|---|---|---|---|---|
| Marginal Distribution Distance | Electricity | 0.615 | 0.368 | 0.135 | 0.117 | 0.236 | 0.006 | 0.027 | 0.001 | 0.001 | 0.020 |
| | Solar | 375.536 | 375.531 | 375.531 | 375.532 | 375.556 | 0.025 | 0.016 | 0.005 | 0.008 | 0.017 |
| | Wind | 0.271 | 0.299 | 0.304 | 0.314 | 0.309 | 0.059 | 0.074 | 0.056 | 0.075 | 0.093 |
| | Traffic | 1.211 | 0.287 | 0.315 | 0.349 | 0.411 | 0.200 | 0.018 | 0.021 | 0.022 | 0.027 |
| | Taxi | 1.008 | 0.433 | 0.338 | 0.371 | 0.384 | 0.154 | 0.013 | 0.003 | 0.020 | 0.019 |
| | Pedestrian | 1.599 | 0.921 | 0.576 | 0.554 | 0.552 | 0.075 | 0.022 | 0.009 | 0.003 | 0.002 |
| | Air | 0.611 | 0.393 | 0.418 | 0.515 | 0.544 | 0.010 | 0.009 | 0.005 | 0.006 | 0.009 |
| | Temperature | 0.487 | 0.317 | 0.356 | 0.330 | 0.307 | 0.113 | 0.025 | 0.020 | 0.017 | 0.042 |
| | Rain | 5.763 | 4.981 | 6.002 | 5.548 | 6.420 | 0.014 | 0.009 | 0.010 | 0.008 | 0.018 |
| | NN5 | 1.550 | 0.796 | 0.613 | 0.616 | 0.573 | 0.130 | 0.118 | 0.004 | 0.005 | 0.009 |
| | Fred-MD | 0.407 | 0.241 | 0.228 | 0.245 | 0.346 | 0.012 | 0.015 | 0.006 | 0.009 | 0.030 |
| | Exchange | 0.365 | 0.359 | 0.376 | 0.309 | 0.330 | 0.062 | 0.063 | 0.067 | 0.083 | 0.046 |

Table 10: Ablation experiment on the impact of the number of prototypes.

| Model | Electricity | Solar | Wind | Traffic | Taxi | Pedestrian |
|---|---|---|---|---|---|---|
| Llama | 0.139 | 375.531 | 0.316 | 0.309 | 0.325 | 0.578 |
| GPT2 | 0.174 | 375.538 | 0.325 | 0.331 | 0.361 | 0.483 |
| | **Air** | **Temperature** | **Rain** | **NN5** | **Fred-MD** | **Exchange** |
| Llama | 0.440 | 0.331 | 5.932 | 0.570 | 0.226 | 0.374 |
| GPT2 | 0.645 | 0.349 | 5.994 | 0.887 | 0.225 | 0.414 |

Table 11: Model performance across different domains. Result measured by MDD

competitive results in both MSE and MAE, even when compared to training on real data. GPT4TS and Time-LLM, on the other hand, exhibit a slight drop in performance when trained on synthetic data, but considerable accepted. In the short-term forecasting scenario (Table 13), the results show similar trends. This suggests that synthetic data can effectively simulate real data patterns, making it a viable option for model training when real-world data is limited or unavailable.

We also compare the quality of our synthetic data generation with the Kernel Synth method employed by Chronos (Ansari et al., 2024). While on the long-term ILI forecasting task models trained on our data clearly outperform models trained on KernelSynth, the picture is slightly more nuanced for shorty-term M4 forecasting. Since the forecast horizons are shorter in M4, the overall difference is less nuanced. Table 14 details for which sub-datasets and models trained on our data are performing better and for which sub-datasets and models KernelSynth is more suitable. These comparisons are further contextualised by statistical analysis - for SMAPE and MASE metrics, we conduct t-test on the individual scores directly. For OWA, since it involves averages of SMAPE and MASE, we use the bootstrapping technique by sampling 1000 times with replacement to obtain the distributions of scores and conduct the t-test on this subset. Overall our method performs better on half model/subset combinations, of which all but two results are statistically significant.

| Methods | LLM4TS | | TEMPO | | Time-LLM | | GPT4TS | |
|---|---|---|---|---|---|---|---|---|
| Metrics | MSE | MAE | MSE | MAE | MSE | MAE | MSE | MAE |
| *ILI-24 KernelSynth* | 4.36 | 1.49 | 1.48 | 1.02 | 1.21 | 0.93 | 3.92 | 1.45 |
| *ILI-36 KernelSynth* | 4.32 | 1.49 | 1.37 | 0.96 | 1.31 | 0.94 | 3.87 | 1.43 |
| *ILI-48 KernelSynth* | 4.15 | 1.48 | 1.69 | 1.09 | 1.49 | 1.04 | 3.77 | 1.40 |
| *ILI-60 KernelSynth* | 4.35 | 1.50 | 2.01 | 1.22 | 1.71 | 1.11 | 3.62 | 1.39 |
| *ILI-24 Ours* | 1.84 | 0.85 | 1.00 | 0.87 | 2.05 | 1.29 | 2.23 | 0.99 |
| *ILI-36 Ours* | 1.86 | 0.86 | 1.22 | 0.99 | 2.13 | 1.34 | 2.13 | 0.97 |
| *ILI-48 Ours* | 1.88 | 0.88 | 1.34 | 1.08 | 2.35 | 1.60 | 2.28 | 1.05 |
| *ILI-60 Ours* | 2.37 | 0.99 | 1.49 | 1.14 | 2.30 | 1.55 | 2.35 | 1.09 |
| *ILI-24 Real* | 1.78 | 0.81 | 0.66 | 0.63 | 1.83 | 1.15 | 1.99 | 0.88 |
| *ILI-36 Real* | 1.75 | 0.82 | 0.92 | 0.80 | 1.90 | 1.17 | 1.90 | 0.90 |
| *ILI-48 Real* | 1.72 | 0.84 | 1.33 | 1.02 | 2.16 | 1.26 | 1.81 | 0.88 |
| *ILI-60 Real* | 2.20 | 0.95 | 0.91 | 0.80 | 2.11 | 1.23 | 1.87 | 0.92 |

Table 12: Comparison of MSE and MAE across various methods on Long-term forecasting. The results are for four different forecasting horizons: $H \in \{24, 36, 48, 60\}$. Red values indicate the best score, and blue values represent the second best.

## N. Prototypes Sample Result

Figure 5 shows 16 semantic prototypes used in our TSG models. Each prototype represents a distinct pattern in time series data, enabling the generation of diverse, domain-specific series. For example, prototypes {0,6,15} capture cyclical patterns useful for seasonal trends. Prototypes {3,12} represent trend patterns, including gradual changes and sharp transitions. Prototypes {1,2,7} show high-frequency fluctuations, representing volatility. By combining these prototypes, the model can generate rich, domain-specific time series data through translating text into time series data with specific semantic concepts. Figure 6 to Figure 17 shows the distribution of prototypes across various domains. Some prototypes, like Prototype 0 in

| Methods | Random | | | LLM4TS | | | TEMPO | | | Time-LLM | | | GPT4TS | | |
|---|---|---|---|---|---|---|---|---|---|---|---|---|---|---|---|
| | SMAPE | MASE | OWA | SMAPE | MASE | OWA | SMAPE | MASE | OWA | SMAPE | MASE | OWA | SMAPE | MASE | OWA |
| *M4-Monthly KernelSynth* | - | - | - | 14.477 | 1.077 | 1.008 | 13.991 | 1.066 | 0.986 | 13.475 | 1.051 | 0.961 | 14.695 | 1.095 | 1.024 |
| *M4-Quarterly KernelSynth* | - | - | - | 12.063 | 1.457 | 1.079 | 11.784 | 1.422 | 1.053 | 11.13 | 1.311 | 0.984 | 11.971 | 1.414 | 1.059 |
| *M4-Yearly KernelSynth* | - | - | - | 16.619 | 3.743 | 0.979 | 16.051 | 3.513 | 0.933 | 13.743 | 3.027 | 0.801 | 17.008 | 3.733 | 0.990 |
| *M4-Monthly Ours* | - | - | - | 13.157 | 0.981 | 0.917 | 12.975 | 0.96 | 0.901 | 13.877 | 1.111 | 1.017 | 13.157 | 0.981 | 0.917 |
| *M4-Quarterly Ours* | - | - | - | 10.608 | 1.253 | 0.939 | 10.318 | 1.207 | 0.909 | 10.877 | 1.342 | 1.022 | 10.608 | 1.253 | 0.939 |
| *M4-Yearly Ours* | - | - | - | 15.547 | 3.72 | 0.944 | 13.466 | 3.036 | 0.794 | 13.788 | 3.255 | 0.843 | 15.547 | 3.72 | 0.944 |
| *Average* | - | - | - | 13.104 | 1.985 | 0.933 | 12.253 | 1.734 | 0.868 | 12.847 | 1.903 | 0.961 | 13.104 | 1.985 | 0.933 |
| *M4-Monthly Real* | 22.756 | 1.959 | 1.71 | 12.817 | 0.947 | 0.890 | 12.698 | 0.934 | 0.879 | 13.327 | 1.023 | 0.943 | 12.916 | 0.958 | 0.898 |
| *M4-Quarterly Real* | 19.216 | 2.587 | 1.816 | 10.301 | 1.207 | 0.908 | 10.077 | 1.177 | 0.887 | 10.672 | 1.266 | 0.946 | 10.386 | 1.230 | 0.920 |
| *M4-Yearly Real* | 37.396 | 8.755 | 2.246 | 13.885 | 3.240 | 0.833 | 13.493 | 3.052 | 0.797 | 13.498 | 3.013 | 0.792 | 14.801 | 3.633 | 0.910 |
| *Average* | 26.456 | 4.434 | 1.924 | 12.334 | 1.798 | 0.877 | 12.089 | 1.721 | 0.854 | 12.499 | 1.767 | 0.894 | 12.701 | 1.940 | 0.909 |

Table 13: Time series forecasting results on unseen time series dataset. The forecasting horizons are in [6, 48] and report value is the average. A lower value indicates better performance. Red: the best, Blue: the second best.

| Frequency | LLM4TS | | | TEMPO | | | GPT4TS | | |
|---|---|---|---|---|---|---|---|---|---|
| | SMAPE | MASE | OWA | SMAPE | MASE | OWA | SMAPE | MASE | OWA |
| Monthly | **True** | **True** | **True** | **True** | **True** | **True** | **True** | **True** | **True** |
| Quarterly | **True** | **True** | **True** | **True** | **True** | **True** | **True** | **True** | **True** |
| Yearly | **True** | **False** | **True** | **True** | **True** | **True** | **True** | **False** | **True** |

Table 14: Statistical significance test results ($p < 0.05$) for differences in SMAPE, MASE and OWA metrics between model trained on our and KernelSynth data based on scores reported in Table 13. Results where our method is better are highlighted in bold.

"temperature" and "electricity," are widely relevant, while others, like Prototype 15 in "traffic," are domain-specific. The sparsity of the heatmaps shows that not all prototypes are equally important within a domain. For example, "solar" primarily uses prototypes {0,6,10,13}. This demonstrates the flexibility of the prototype-based approach, capturing both general and domain-specific patterns.

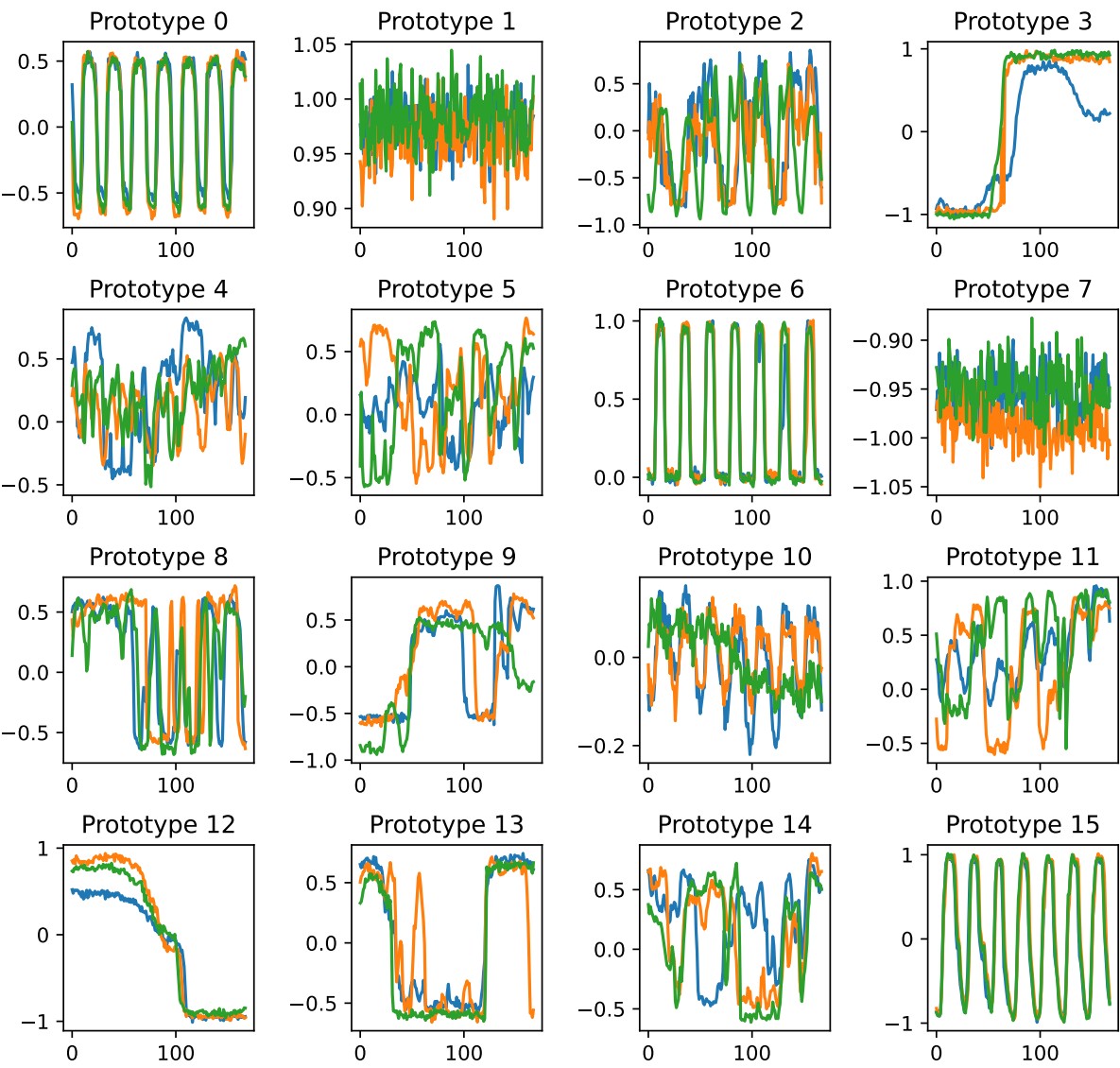

Figure 5: Visualization of semantic prototypes. Each prototype represents a different pattern or characteristic commonly found in time series data.

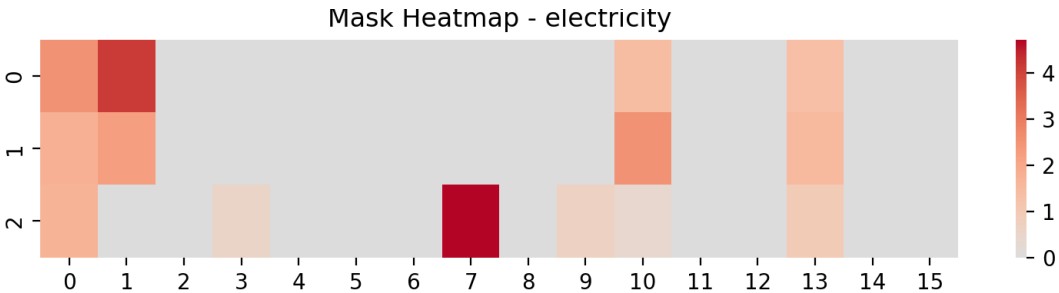

Figure 6: Prototype distribution for the electricity domain. Each heatmap shows prototype indices (x-axis, 0–15) and their frequency or importance (color intensity).

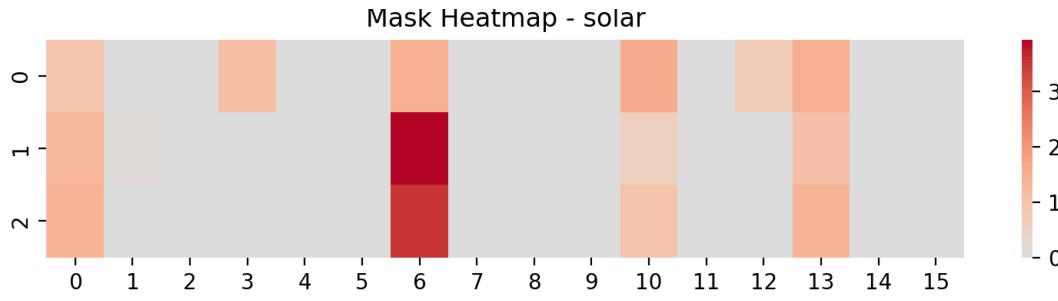

Figure 7: Prototype distribution for the solar domain. Each heatmap shows prototype indices (x-axis, 0–15) and their frequency or importance (color intensity).

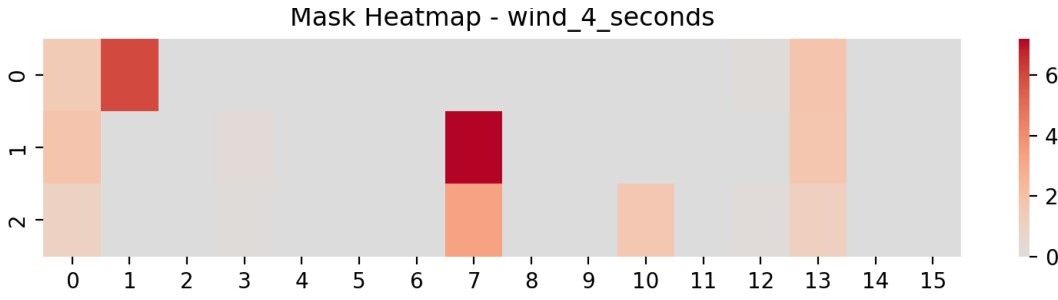

Figure 8: Prototype distribution for the wind (4 seconds) domain. Each heatmap shows prototype indices (x-axis, 0–15) and their frequency or importance (color intensity).

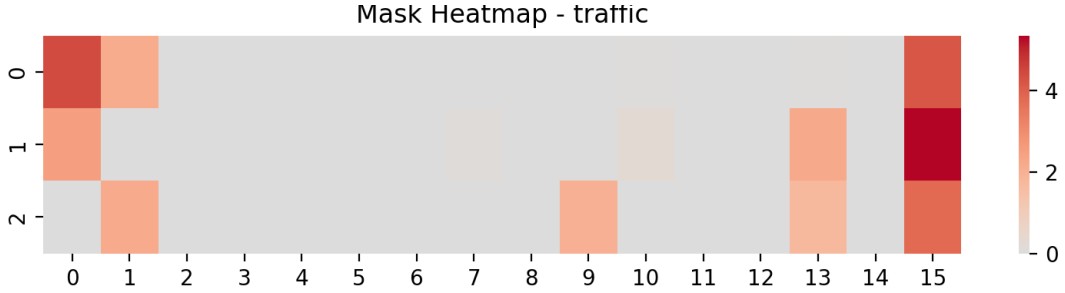

Figure 9: Prototype distribution for the traffic domain. Each heatmap shows prototype indices (x-axis, 0–15) and their frequency or importance (color intensity).

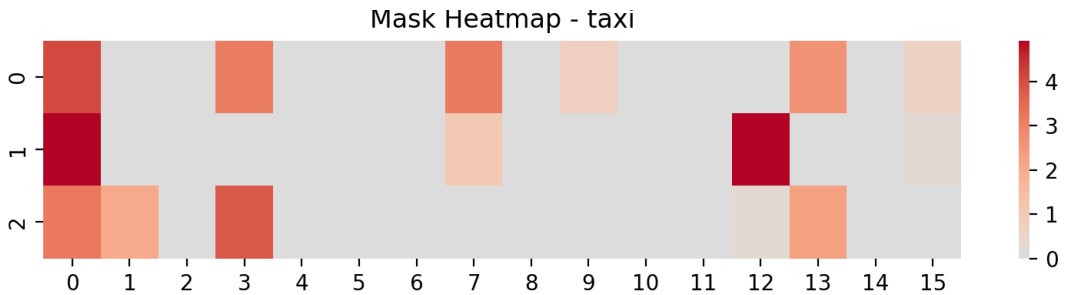

Figure 10: Prototype distribution for the taxi domain. Each heatmap shows prototype indices (x-axis, 0–15) and their frequency or importance (color intensity).

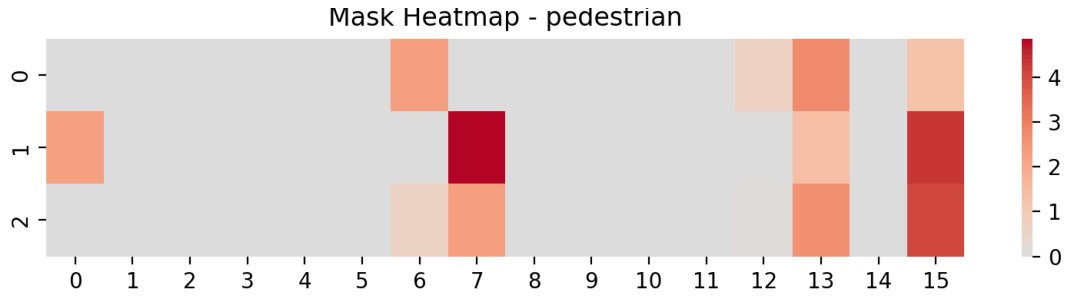

Figure 11: Prototype distribution for the pedestrian domain. Each heatmap shows prototype indices (x-axis, 0–15) and their frequency or importance (color intensity).

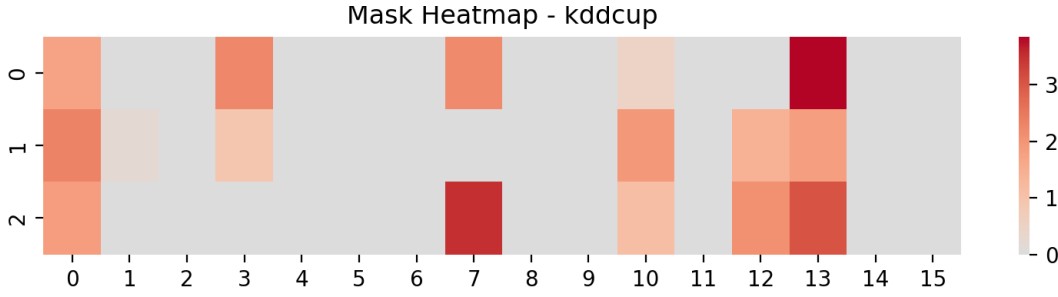

Figure 12: Prototype distribution for the kddcup domain. Each heatmap shows prototype indices (x-axis, 0–15) and their frequency or importance (color intensity).

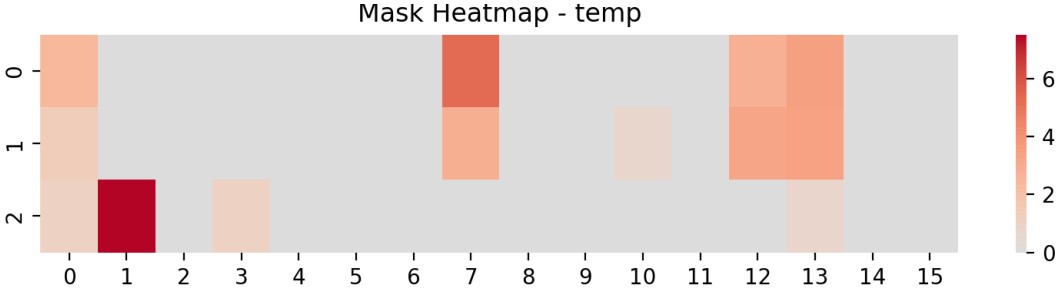

Figure 13: Prototype distribution for the temperature domain. Each heatmap shows prototype indices (x-axis, 0–15) and their frequency or importance (color intensity).

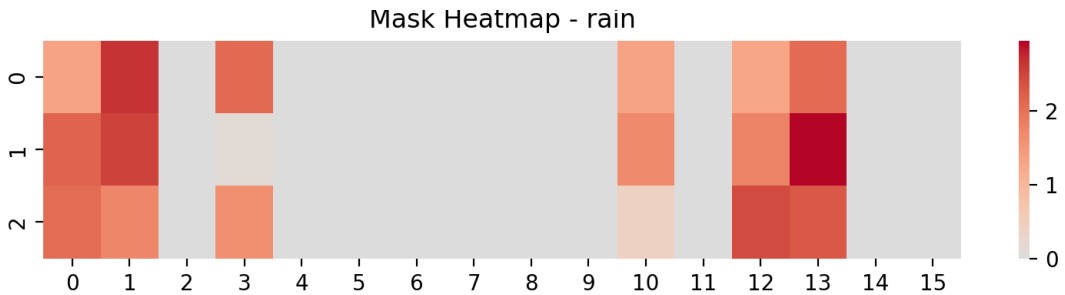

Figure 14: Prototype distribution for the rain domain. Each heatmap shows prototype indices (x-axis, 0–15) and their frequency or importance (color intensity).

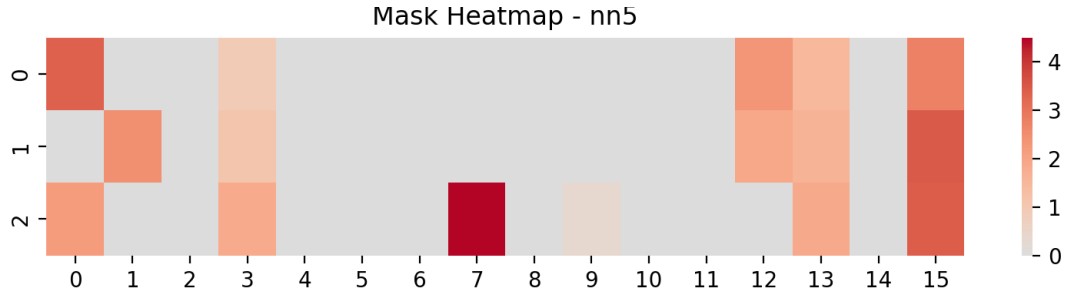

Figure 15: Prototype distribution for the nn5 domain. Each heatmap shows prototype indices (x-axis, 0–15) and their frequency or importance (color intensity).

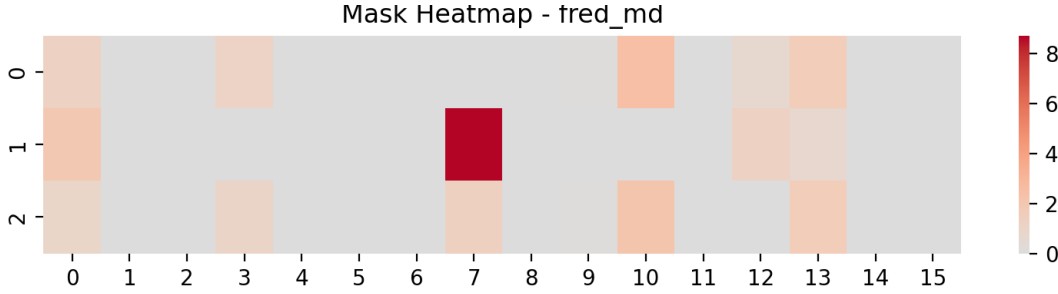

Figure 16: Prototype distribution for the fred_md domain. Each heatmap shows prototype indices (x-axis, 0–15) and their frequency or importance (color intensity).

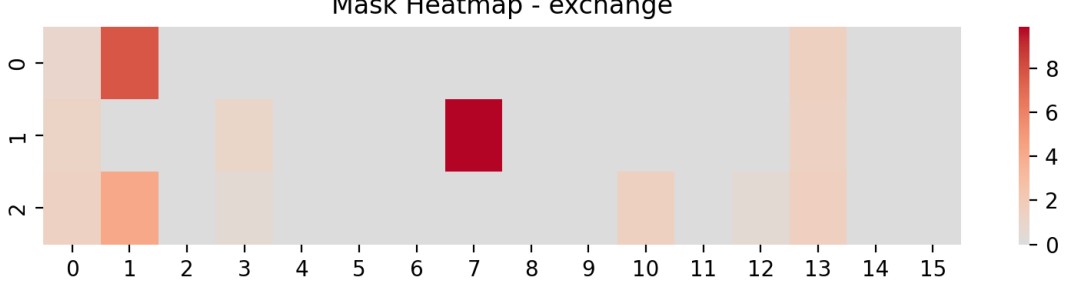

Figure 17: Prototype distribution for the exchange domain. Each heatmap shows prototype indices (x-axis, 0–15) and their frequency or importance (color intensity).

