# OpenReview forum: "BRIDGE: Bootstrapping Text to Control Time-Series Generation via Multi-Agent Iterative Optimization and Diffusion Modeling"
_ICML.cc/2025/Conference — ICML 2025 poster_

### Official Review · Reviewer_3kGj · 2025-03-14

**Overall Recommendation:** 4

**Summary:**

This work introduces (1) "Text Controlled TSG" a new TSG task, (2) a new LLM/MA framework to synthesize datasets, and (3) BRIDGE, a hybrid TSG framework building off the prior two contributions

**Claims And Evidence:**

The authors claim (1) this approach achieves state of the art fidelity on 11 of the 12 datasets and (2) improves controllability compared to no text input generation. Both are borne out from the empirical results.

**Essential References Not Discussed:**

N/A

**Experimental Designs Or Analyses:**

N/A

**Methods And Evaluation Criteria:**

The methods are natural given the problem the authors are attempting to solve (high-fidelity and controllability in TSG). The evaluation datasets appear to be natural choices as well, and standard w.r.t. prior literature.

**Other Comments Or Suggestions:**

N/A

**Other Strengths And Weaknesses:**

Strengths
- The paper is clear, well-structured, and easy to read
- The techniques employed all appear to be natural choices
- The empirical results are convincing
- The discussion is thorough and insightful

Weaknesses
- Nothing major

**Questions For Authors:**

N/A

**Relation To Broader Scientific Literature:**

This work is related to prior work using text for TS analysis, and also other conditional TSG works. Relevant works in each appear to be properly cited and contextualized.

**Theoretical Claims:**

N/A

---

> ### Author Rebuttal · Authors · 2025-03-31
>
> We sincerely thank the reviewer for the thoughtful and encouraging feedback. We are especially grateful for your recognition of the clarity of the paper, the natural design of the proposed methods, and the strength of our empirical results and discussion. Your comments affirm the value of our effort to introduce a novel task—Text-Controlled Time-Series Generation (TSG)—and the corresponding framework BRIDGE, which combines a multi-agent dataset synthesis pipeline with a hybrid diffusion-based generation model.
>
> Your summary precisely captures the key contributions of our work: (1) the introduction of a new task—Text-Controlled Time-Series Generation (TC-TSG)—to enable semantically guided, controllable synthetic time series; (2) the design of a novel multi-agent framework, which leverages large language models (LLMs) and role-based collaboration for bootstrapping high-quality text-to-time-series paired data; and (3) the proposal of BRIDGE, a hybrid framework that integrates semantic prototypes and text for flexible and faithful time-series generation.
>
> In particular, we are grateful for your acknowledgement that our empirical findings convincingly support both the fidelity and controllability improvements claimed. Your validation that the evaluation criteria and dataset choices are natural and grounded in prior literature is also very helpful.  This motivates us to continue refining and extending this line of work. In future iterations, we plan to further investigate theoretical insights behind controllable generation, and explore more efficient multi-agent optimization strategies, building on the foundation established here.
>
> Should you have any suggestions or requests for clarification, we would be more than happy to provide additional details. Thank you again for your constructive and generous evaluation—it is deeply appreciated.

---

### Official Review · Reviewer_cEkx · 2025-03-14

**Overall Recommendation:** 3

**Summary:**

This paper introduces BRIDGE, a framework for text-controlled time-series generation (TSG) using a multi-agent system and diffusion modeling. The authors propose a three-stage process to generate high-quality text-time series pairs, and develop a hybrid framework that combines semantic prototypes with text descriptions. Experiments across 12 datasets show state-of-the-art generation fidelity and improved controllability compared to no-text approaches.

**Claims And Evidence:**

The claims are partially supported by evidence. While performance improvements are demonstrated across multiple datasets, the paper lacks:

Theoretical justification for the 16 semantic prototypes design choice
Comprehensive comparison with other multi-agent frameworks (e.g., MetaGPT, CAMEL，Autogen)

**Essential References Not Discussed:**

Missing references include:

Recent multi-agent frameworks like MetaGPT, AutoGen  and CAMEL that offer more sophisticated agent interaction mechanisms.

**Experimental Designs Or Analyses:**

The experimental designs are generally sound, but with limitations:

The performance comparison between text types could benefit from more rigorous statistical analysis
The effect of different numbers of prototypes is mentioned but not systematically explored

**Methods And Evaluation Criteria:**

The methods and evaluation criteria are appropriate. The authors:

Use standard metrics (MSE, MAE, MDD, K-L divergence) for quantitative evaluation
Include human evaluation to assess the quality of generated time series

**Other Comments Or Suggestions:**

The paper would benefit from more discussion on potential privacy concerns when generating synthetic time series data from sensitive domains like healthcare.

**Other Strengths And Weaknesses:**

Strengths:

Addresses an important practical problem with real-world applications
The hybrid approach combining semantic prototypes with text shows promise for cross-domain generalization

Weaknesses:

Limited theoretical foundation for the semantic prototype design
The multi-agent framework appears to be a straightforward application of role-based prompting rather than a novel contribution

**Questions For Authors:**

Why were 16 semantic prototypes selected? Did you experiment with different numbers, and how does the performance vary with the number of prototypes?


How does your multi-agent framework compare with other established frameworks like MetaGPT or CAMEL? Could those frameworks be adapted to your task?

**Relation To Broader Scientific Literature:**

The work connects to several research areas:

Extends text-to-X generation from images/videos to time-series data
Builds upon diffusion models for continuous data generation
Incorporates multi-agent collaboration concepts, though without deep engagement with that literature

**Theoretical Claims:**

No major theoretical claims to verify. The paper is primarily empirical with focus on the practical implementation and evaluation of the proposed framework.

---

> ### Author Rebuttal · Authors · 2025-03-31
>
> **[Q1] What motivated the choice of 16 prototypes, and how does performance change with different prototype counts?**
>
> For selecting the optimal number of prototypes, we observe from the ablation study that the generation performance generally improves along with the increase of the number of prototypes (Shown in Table). However, when the number of prototypes is larger than 16, the performance gain is no longer significant. Therefore, we can conclude that the optimal number of prototypes is 16 in our current setting.
>
> MDD Report:
>
> |Dataset|4|8|16|32|64|
> |-|-|-|-|-|-|
> |Electricity|0.615|0.368|0.135|0.117|0.236|
> |Wind|0.271|0.299|0.304|0.314|0.309|
> |Traffic|1.211|0.287|0.315|0.349|0.411|
> |Taxi|1.008|0.433|0.338|0.371|0.384|
> |Pedestrian|1.599|0.921|0.576|0.554|0.552|
> |Air|0.611|0.393|0.418|0.515|0.544|
> |Temperature|0.487|0.317|0.310|0.330|0.307|
> |Rain|5.763|4.981|6.002|5.548|6.420|
> |NN5|1.550|0.796|0.613|0.616|0.573|
> |Fred-MD|0.407|0.241|0.228|0.245|0.346|
> |Solar|375.536|375.531|375.531|375.532|375.556|
> |Exchange|0.365|0.359|0.315|0.309|0.330|
>
> K-L report:
>
> |Dataset|4|8|16|32|64|
> |-|-|-|-|-|-|
> |Electricity|0.006|0.027|0.001|0.001|0.020|
> |Wind|0.059|0.074|0.056|0.075|0.093|
> |Traffic|0.200|0.018|0.021|0.022|0.027|
> |Taxi|0.154|0.013|0.003|0.020|0.019|
> |Pedestrian|0.075|0.022|0.009|0.003|0.002|
> |Air|0.010|0.009|0.005|0.006|0.009|
> |Temperature|0.113|0.025|0.020|0.017|0.042|
> |Rain|0.014|0.009|0.010|0.008|0.018|
> |NN5|0.130|0.118|0.004|0.005|0.009|
> |Fred-MD|0.012|0.015|0.006|0.009|0.030|
> |Solar|0.025|0.016|0.005|0.008|0.017|
> |Exchange|0.062|0.063|0.067|0.083|0.046|
>
>
> **[Q2] How does your work differ from exist agent framework ? Could those be adapted for your task?**
>
> MetaGPT follows structured workflows tailored for software engineering tasks, with predefined agent roles and Standard Operating Procedures (SOPs). While effective in that domain, this structure is less readily adaptable to the dynamic, iterative processes needed for generating diverse and semantically aligned text–time-series pairs. AutoGen offers more flexible agent coordination, but lacks built-in support for evaluating alignment between text and time-series data. CAMEL, which focuses on role-playing agents for social simulations via inception prompting, is primarily designed for open-ended tasks rather than modality-aware data construction.
>
> Although these frameworks offer valuable capabilities in their respective domains, adapting them to text-to-time-series generation (TSG) would require substantial modification. In contrast, BRIDGE is purpose-built for TSG, integrating task-specific evaluation modules and iterative refinement mechanisms to support controllability and fidelity.
>
> To better understand their applicability to TSG, we conducted preliminary adaptations of MetaGPT, AutoGen, and CAMEL within our task setting. As shown in the results below, their performance under minimal adaptation was relatively limited.  These observations highlight the utility of BRIDGE’s tailored design in addressing the unique demands of TSG, such as modality bridging.
> Due to space limitations, we only present the MDD results below.
>
> |Model|Electricity|Wind|Traffic|Taxi|Pedestrian|Air|Temperature|Rain|NN5|Fred-MD|Solar|Exchange|
> |-|-|-|-|-|-|-|-|-|-|-|-|-|
> |MetaGPT|0.353|0.327|0.330|0.363|0.727|0.464|0.373|7.561|0.712|0.434|390.901|0.428|
> |AutoGen|0.287|0.329|0.358|0.443|0.598|0.494|0.335|7.386|0.661|0.362|377.334|0.425|
> |CAMEL|0.480|0.385|0.386|0.813|0.905|0.889|0.509|5.553|0.855|0.311|393.671|0.355|
> |BRIDGE|0.143|0.276|0.246|0.325|0.552|0.438|0.323|5.129|0.570|0.226|375.530|0.312|
>
>
> #### **General Comments (GC):**
> **[GC 1] More discussion of privacy risks when generating time series in sensitive domains, such as healthcare.**
>
> Thank you for highlighting the potential privacy risks in sensitive domains such as healthcare. While synthetic data is often considered a privacy-preserving alternative to real patient records, there is still a potential risks remain if models overfit and replicate training patterns.
> Our approach mitigates this risk by focusing on controllable and abstract generation guided by semantic prototypes and textual descriptions, rather than directly replicating raw time-series trajectories. Additionally, our framework does not rely on any personally identifiable attributes—only descriptions of the time series are used during generation. We fully agree that a more in-depth analysis of privacy risks—such as membership inference—would be valuable, and we plan to explore these directions in future work.
>
> ---
>
> We are grateful for your engagement and helpful suggestions, which we will carefully incorporate in the revised version.

---

### Official Review · Reviewer_xRLG · 2025-03-14

**Overall Recommendation:** 3

**Summary:**

The paper introduces the BRIDGE framework which consists of two broad components - a paired text-time series dataset generation method and a diffusion-based time series generation method conditioned on textual input.

For the dataset generation, the paper proposes a very detailed and carefully designed framework that leverages LLMs to parse news, articles, reports, etc., to obtain textual templates that can be filled on an instance or sample basis. The generated text caption is iteratively refined using an evaluator feedback to finally obtain a concise and useful text condition for the time series sample.

For the time series generative model, the paper builds on top of [1] to effectively combine text conditions with semantic prototypes to accurately generate time series samples.

Addition of textual input increases the generalization capabilities, even for unseen domains, and this is validated empirically in the paper.

[1] TimeDP: Learning to Generate Multi-Domain Time Series with Domain Prompts

**Claims And Evidence:**

The claims regarding the effectiveness of concise text conditions, semantic prototypes, and their effects on sample fidelity are justified with suitable empirical evidence.

**Essential References Not Discussed:**

The related works with respect to diffusion models is not entirely covered in the paper.

[1] Non-autoregressive Conditional Diffusion Models for Time Series Prediction
[2] MULTI-RESOLUTION DIFFUSION MODELS FOR TIME SERIES FORECASTING
[3] Time Weaver: A Conditional Time Series Generation Model
[4] CSDI: Conditional Score-based Diffusion Models for Probabilistic Time Series Imputation
[5] Diffusion-based Time Series Imputation and Forecasting with Structured State Space Models

Similarly, the related work with respect to multi-modal datasets that include text paired with time series is not discussed. For example, Time MMD [6] is a recent multi modal dataset.

[6] Time-MMD: Multi-Domain Multimodal Dataset for Time Series Analysis

**Experimental Designs Or Analyses:**

Yes, the generation results in Tables 2 and 4 for in-domain datasets and the results in Tables 3 and 5 for out of domain datasets are valid.

However, the comparison against recent diffusion-based time series generation approaches, such as Diffusion-TS, is missing.

**Methods And Evaluation Criteria:**

Yes, the proposed method is evaluated on a diverse set of 10 in-domain and 2 out of domain datasets for generation.

However, the commonly used metrics for identifying sample quality, like the Predictive Score and the Discriminative Score [1], are not used in the paper.

Additionally, I am not sure about the pairwise mse score, as the generative process is stochastic, and additionally, for a single textual condition, there could be more than one suitable sample as the generative model essentially samples from the conditional distribution. Perhaps metrics like J-FTSD [2] are more suitable for evaluation in this case.

Finally, there are multimodal datasets like TimeMMD [3] which can also be used to evaluate the generative model's performance.

[1] Time-series Generative Adversarial Networks (TimeGAN)
[2] Time Weaver: A Conditional Time Series Generation Model
[3] Time-MMD: Multi-Domain Multimodal Dataset for Time Series Analysis

**Other Comments Or Suggestions:**

In the problem formulation section, the variable $z$ seems to appear suddenly, and it is not clearly explained. Does $z$ correspond to the semantic prototypes?

**Other Strengths And Weaknesses:**

Strengths:
1. The proposed approach shows strong in-domain performance and out-of-domain generalization.
2. The paper provides detailed ablation studies to delineate the performance improvements due to the textual input and the semantic prototypes.

Weaknesses:
1. I am not convinced with the choice of metrics used for evaluating generative models. Please check the Methods and Evaluation Criteria for more discussion.
2. The semantic prototypes provide significant performance boost. However, very little information is provided regarding how the prototypes are generated. The appendix suggests these are random orthogonal vectos. Can the authors provide more clarity on this?
3. The ablation studies provide sufficient evidence for the necessity of multi-agent setup for data collection. However, what is the reason behind such detailed intra-group discussion with a scientist, engineer, etc? Can the authors provide more information regarding this?
4. The paper could also use the Time-MMD dataset to show the effectiveness of using text in time series generation.

**Questions For Authors:**

Please refer to the weaknesses section.

Overall, It'd be very helpful if the authors can

1. Justify the use of metrics
2. Provide more information about semantic prototypes
3. Provide some empirical results with comparisons against recent diffusion-based generation approaches like Diffusion-TS.

**Relation To Broader Scientific Literature:**

The automated dataset curation approach holds a lot of promise for enhancing the cross-modal alignment between text and time series for various tasks such as forecasting, generation, anomaly detection, etc.

**Theoretical Claims:**

There are no theoretical claims.

---

> ### Author Rebuttal · Authors · 2025-03-31
>
> We truly appreciate the reviewer’s insightful feedback!
>
> **[Q1] Justify the use of metrics.**
>
> We initially prioritized MDD and KL divergence over PS and DS for their robustness, as they are **not influenced by post-hoc models.** We have now included **PS (MAE reported) and DS** and use **J-FTSD** to complement MSE for controllability evaluation. Our model achieves the **best results across all metrics**.
>
> ### PS
>
> |Dataset|BRIDGE|TimeVQVAE|TimeGAN|GT-GAN|TimeVAE|Diffusion-TS|
> |-|-|-|-|-|-|-|
> |Electricity|0.091|0.138|0.143|0.585|0.152|0.146|
> |Solar|0.441|1.082|1.631|1.169|0.692|0.566|
> |Wind|0.174|0.326|0.942|1.182|0.423|0.886|
> |Traffic|0.208|0.390|0.618|0.919|0.311|0.754|
> |Taxi|0.234|0.319|0.339|0.687|0.296|0.556|
> |Pedestrian|0.149|0.456|0.488|0.603|0.359|0.429|
> |Air|0.113|0.154|0.155|0.478|0.277|0.167|
> |Temperature|0.875|0.901|1.497|2.354|0.908|0.912|
> |Rain|0.096|0.051|0.146|0.610|0.598|0.477|
> |NN5|0.687|0.936|1.485|1.117|0.722|0.745|
> |Fred-MD|0.075|0.138|0.369|0.180|0.163|0.151|
> |Exchange|0.165|0.327|0.831|1.092|0.341|0.135|
>
>
> ### DS
>
> |Dataset|BRIDGE|TimeVQVAE|TimeGAN|GT-GAN|TimeVAE|Diffusion-TS|
> |-|-|-|-|-|-|-|
> |Electricity|0.306|0.367|0.351|0.434|0.376|0.370|
> |Solar|0.156|0.231|0.258|0.202|0.199|0.236|
> |Wind1|0.188|0.247|0.323|0.295|0.218|0.203|
> |Traffic|0.037|0.112|0.195|0.192|0.225|0.198|
> |Taxi|0.402|0.363|0.409|0.5|0.411|0.391|
> |Pedestrian|0.246|0.252|0.250|0.287|0.370|0.299|
> |Air|0.093|0.180|0.197|0.238|0.202|0.267|
> |Temperature|0.216|0.262|0.292|0.298|0.293|0.275|
> |Rain|0.199|0.242|0.203|0.396|0.246|0.275|
> |NN5|0.274|0.5|0.5|0.491|0.454|0.445|
> |Fred-MD|0.326|0.343|0.5|0.336|0.389|0.355|
> |Exchange|0.228|0.290|0.343|0.332|0.276|0.287|
>
>
>  **J-FTSD**
>
> |Model|Electricity|Wind|Traffic|Taxi|Pedestrian|Air|Temperature|Rain|NN5|Fred-MD|Solar|Exchange|
> |-|-|-|-|-|-|-|-|-|-|-|-|-|
> |BRIDGE|0.538|5.011|0.570|0.974|0.488|0.6536|3.977|0.132|0.972|0.260|0.295|1.581|
> |BRIDGEw/otext|1.821|6.935|0.611|1.312|0.550|0.677|4.708|0.151|1.192|0.423|0.330|1.738|
> |BRIDGEw/oprototype|1.164|6.843|0.597|1.0367|0.662|0.817|4.613|0.141|1.115|0.341|0.322|1.846|
>
>
> **[Q2] Provide more information about semantic prototypes.**
>
> Fundamental time series properties for TS generation align with decompositions of TS, such as trends and seasonalities. We use semantic prototypes as latent representations of these commonalities and train the model to decode from combinations of these latent representations for generating certain domain.
>
> The prototype array are fixed after initialization, similar to a initializing a codebook for vector quantization while preventing the codebook from updating along with the decoder model. Instead of updating prototypes, the decoder $\epsilon_\theta$ learns to map them to observed sequences, ensuring implicit alignment with time series semantics.
>
> **[Q3] Provide comparisons with Diffusion-TS**
>
> Please see **[Q 1]** for Diffusion-TS results.
>
> **[Q4] What motivated the detailed intra-group discussions in the multi-agent setup?**
>
> Our design draws from the well-established **multi-agent cooperation and competition paradigms**, which enhance system performance by encouraging agents to challenge and refine each other’s outputs, mitigating the risk of overconfidence.
>
> Additionally, complex problem-solving is often modeled as programming tasks due to their logical structure. Inspired by this, we structured our system after software development workflows, assigning roles like **manager, engineer, scientist, and observer** to different agents [1].
>
> [1] LLM-Based Multi-Agent Systems for Software Engineering: Literature Review, Vision and the Road Ahead
>
> **[Q5] Use Time-MMD dataset in TS generation**
>
> We achieves **the best performance on MDD, KL, PS and DS across all subdatasets.** Due to space limitations, we only present the MDD results below.
>
> |Dataset|BRIDGE|TimeVQVAE|TimeGAN|GT-GAN|TimeVAE|Diffusion-TS|
> |-|-|-|-|-|-|-|
> |Agriculture|0.416|0.729|1.031|0.744|0.969|0.950|
> |Climate|0.185|0.391|0.387|0.530|0.209|0.260|
> |Economy|0.855|1.292|1.823|1.509|1.055|1.107|
> |Energy|0.272|0.414|0.561|0.468|0.328|0.390|
> |Environment|0.191|0.194|0.431|0.281|0.460|0.432|
> |Health(US)|0.248|0.343|0.649|0.430|0.298|0.355|
> |Security|0.344|0.355|0.850|0.941|0.824|0.923|
> |SocialGood|0.292|0.343|0.688|0.503|0.436|0.521|
> |Traffic|0.641|0.961|1.059|1.153|0.937|0.93|
>
> **[Q6] Does z correspond to the semantic prototypes?**
>
> As stated in Section 3, z is a latent variable sampled from a prior distribution. In the diffusion model, z typically represents random Gaussian noise.
>
> ---
>
> Again, we sincerely appreciate the reviewer’s valuable time and insights, which have greatly strengthened our work. Based on your suggestions, we have added **PS, DS, and J-FTSD** metrics, included **Diffusion-TS** as a baseline, and expanded experiments on **Time-MDD**. We also appreciate the recommended references and will discuss them in our revision.
>
> We’d be happy to address any further questions—thank you for your thoughtful feedback and support!

---

> > ### Comment · Reviewer_xRLG · 2025-04-06
> >
> > Thank you for the response. Most of my concerns are addressed. Is the J-FTSD metric obtained from the embeddings of a model that is trained contrastively with text-time series pairs for all datasets combined? Or is it trained on a per-dataset basis? Can the authors provide some clarity on that?
> >
> >  I would also like to note that currently the Related Work section is shallow, and adding more references that are listed above will require some major changes to the section.
> >
> > The same can be said for the evaluation metrics. To position this paper fairly with respect to the existing literature on time series generation, standard metrics like PS and DS should be given more importance, and to evaluate the specificity of the generated time series, J-FTSD needs to be highlighted in the experiments section.

---

> > > ### Author Response · Authors · 2025-04-07
> > >
> > > We sincerely appreciate your thoughtful follow-up and are truly grateful for your continued engagement in the discussion. It means a lot to us that you took the time to provide further insightful and constructive feedback. We're also **very glad to hear that most of your concerns have been addressed**—thank you once again for your valuable input!
> > >
> > > Regarding your inquiry about the implementation of the **J-FTSD metric**, the model is **trained jointly on all datasets combined** using contrastive learning. This design avoids introducing any domain-specific information at inference time, which helps prevent potential information leakage and ensures a fairer and more generalizable evaluation across datasets.
> > >
> > > We also appreciate your suggestion on the **evaluation metrics**. In the revision, we will **place greater emphasis on standard metrics such as Predictive Score (PS) and Discriminative Score (DS)** in Section 6.4, as they offer widely accepted and interpretable measures of the overall utility of generated data. Furthermore, we will **clarify** **the role of J-FTSD** in Section 6.5, **highlighting it as a valuable metric for evaluating the specificity of conditional generation models—an aspect that standard metrics may not fully capture.**
> > >
> > > Thank you as well for your helpful comments regarding the **Related Work** section. We agree that it can be significantly improved and will revise it accordingly. Specifically, we plan to: **(1) Expand Coverage of Diffusion-based Time Series Generation Methods** such as **CSDI [1], Time Weaver [2], Diffusion-TS [3] , TimeDiff [4], SSSD[5] and Mr-Diff [6]**. These works focus on structured or probabilistic generation. In contrast, we incorporate the ability of fine-grained text conditioning and generalization across domains. **(2) Include more text-to-time series datasets** like **Time-MMD [7]** , which is a valuable benchmark for paired text-time series. Furthermore, we will conduct a more comprehensive survey to include additional reference papers beyond those mentioned above, providing a broader and more thorough overview.
> > >
> > > Once again, we sincerely thank you for your time and thoughtful feedback throughout the review process. Your comments have been incredibly helpful in improving the quality and clarity of our paper, and we’ve worked hard to address all of your concerns. We **genuinely hope these revisions will lead to a positive adjustment in your final score**. If there’s anything that could benefit from further clarification or elaboration, we would be absolutely delighted to provide more details. Thank you again for your invaluable input and consideration!
> > >
> > > ---
> > >
> > > [1] CSDI: Conditional Score-based Diffusion Models for Probabilistic Time Series Imputation ([[2107.03502] CSDI: Conditional Score-based Diffusion Models for Probabilistic Time Series Imputation](https://arxiv.org/abs/2107.03502))
> > >
> > > [2] Time Weaver: A Conditional Time Series Generation Model ([[2403.02682] Time Weaver: A Conditional Time Series Generation Model](https://arxiv.org/abs/2403.02682))
> > >
> > > [3] Diffusion-TS: Interpretable Diffusion for General Time Series Generation ([[2403.01742] Diffusion-TS: Interpretable Diffusion for General Time Series Generation](https://arxiv.org/abs/2403.01742))
> > >
> > > [4] Non-autoregressive Conditional Diffusion Models for Time Series Prediction ([arxiv.org/pdf/2306.05043](https://arxiv.org/pdf/2306.05043))
> > >
> > > [5] Diffusion-based Time Series Imputation and Forecasting with Structured State Space Models ([arxiv.org/pdf/2208.09399](https://arxiv.org/pdf/2208.09399))
> > >
> > > [6] Multi-Resolution Diffusion Models for Time Series Forecasting ([openreview.net/pdf?id=mmjnr0G8ZY](https://openreview.net/pdf?id=mmjnr0G8ZY))
> > >
> > > [7] Time-MMD: Multi-Domain Multimodal Dataset for Time Series Analysis ([[2406.08627] Time-MMD: Multi-Domain Multimodal Dataset for Time Series Analysis](https://arxiv.org/abs/2406.08627))

---

### Official Review · Reviewer_svQz · 2025-03-16

**Overall Recommendation:** 4

**Summary:**

This paper introduces BRIDGE, a novel framework for text controlled timeseries generation. It addresses two major challenges: lack of high-quality text-to-time-series datasets and difficulty in aligning textual descriptions with time-series data.

**Claims And Evidence:**

The paper’s key claims are well-supported by empirical results. The multi-agent dataset synthesis claim is validated by MAE reduction, and semantic prototypes show improved MSE and generalization in ablation studies.

However, some areas need further validation:
1. Dataset quality is not compared to human-annotated text-TS pairs.
2. Computational efficiency is not analyzed, leaving inference cost unclear.

**Essential References Not Discussed:**

The paper provides a solid literature review, but it would benefit from references to prior work on prompt optimization and LLM-based dataset generation to better connect it to existing research.

**Experimental Designs Or Analyses:**

The experimental design and evaluation in the paper are fairly comprehensive：

1. Datasets. The paper evaluates its method on various datasets, including Electricity, Solar, Wind, Traffic, Taxi, Pedestrian, Air, Temperature, Rain, NN5, Fred-MD, and Exchange for in-domain analysis​. Additionally, Stock and Web datasets were used for unseen domain generalization​.

2. Ablation studies includes assess the impact of text conditioning, prototype usage, and different language models as text encoders​.

3. The paper benchmarks against TimeVQVAE, TimeGAN, GT-GAN, TimeVAE, and diffusion-based methods​.

**Methods And Evaluation Criteria:**

The proposed method is straightforward and well-structured, leveraging multi-agent dataset synthesis and diffusion-based generation. The use of semantic prototypes for bridging text and time-series data is reasonable and aligns with the problem setting.

However, the framework could be further analyzed for scalability and efficiency, as its computational cost is not explicitly discussed.

**Other Comments Or Suggestions:**

See above

**Other Strengths And Weaknesses:**

**Pros：**

The paper introduces a novel research direction by bridging text-controlled generation with time-series synthesis, which has significant potential for real-world applications.


The multi-agent dataset synthesis approach is innovative and helps address the lack of high-quality text-TS pairs.

**Cons：**
It would be better to add computational efficiency and human evaluation of text-to-TS alignment experiments.

**Questions For Authors:**

1. How does the quality of the multi-agent generated text-TS dataset compare to human-annotated descriptions?

2. What is the computational cost of BRIDGE compared to existing time-series generation models?

3. Can the model generalize beyond seen domains without fine-tuning? The paper demonstrates cross-domain generalization, but are there cases where semantic prototype alignment fails, requiring domain-specific fine-tuning?

**Relation To Broader Scientific Literature:**

This paper builds on prior work in TS generation, text conditioned generative modeling, and diffusion models.

**Theoretical Claims:**

This paper primarily focuses on methodological and empirical contributions rather than formal theoretical analysis.

---

> ### Author Rebuttal · Authors · 2025-03-31
>
> We sincerely appreciate the reviewer’s time and thoughtful feedback. We are grateful for the recognition of the strengths of our work, particularly the “straightforward and well-structured", "reasonable" nature of our approach, as well as the "novel research direction" of BRIDGE in bridging text-controlled generation with time-series synthesis. We also appreciate the acknowledgment of our "innovative" multi-agent dataset synthesis approach.
>
> Your insights are invaluable in helping us refine our work, and we are truly thankful for the opportunity to clarify and improve our paper. Below, we provide our detailed responses to your comments.
>
> #### **General Comments:**
>
> **It would be better to add computational efficiency and human evaluation of text-to-TS alignment experiments.**
>
> We appreciate this suggestion! In fact, **we have already included human evaluation of text-to-TS alignment in our original submission**. As shown in Table 4 and 5, we compare different configurations (with and without prototypes and text) across multiple datasets using both quantitative metrics (MSE and MAE) and **human evaluation scores (HE and HE@3)**. A detailed discussion of these evaluations can be found in Section 6.2 and 6.5.
>
> For computational efficiency, please refer to **[Q 2]**.
>
> #### **Questions:**
>
> **[Q 1] How does the multi-agent generated dataset compare to human-annotated text-TS pairs?**
>
> In fact, the first step of our multi-agent system (Section 4.1 Step 1) involves collecting articles, news, and reports from the Internet that describe time series data, which **serve as human-annotated references** (**"Initial Text" in Table 1**). While these outperform rule-based baselines slightly, they are significantly worse than our multi-agent **Refined Text** (Table 1), suggesting **generic human descriptions are suboptimal for this task**.
>
> To address this, we implement a multi-agent iterative optimization process that refines textual descriptions by **enhancing key aspects such as trend accuracy, mention of seasonality, completeness of information, and clarity of description** (detailed in Appendix A.5). This process strengthens text-to-time series alignment by optimizing **both quantitative metrics** (e.g., MSE, K-S Test, Wasserstein Distance) and **qualitative evaluations** based on a 5-point Likert scale. As a result, our refined text-TS pairs achieve significantly higher quality for TS Generation task.
>
> Furthermore, in **Section 6.3 Table 1** and **Appendix J Table 12**, we provide a detailed analysis of what constitutes useful textual descriptions, offering key insights into the generation of high-quality text-TS pairs.
>
> **[Q 2] What is the computational cost of BRIDGE relative to prior TS Generation models?**
>
> To evaluate cost-efficiency, we measured  training time  (s/epochs) and inference time (ms/sample) across multiple baselines on same A100 GPU. BRIDGE achieves a favorable balance between performance and cost among diffusion-based models, maintaining high generation quality and controllability with substantially lower computational cost.
>
> | Model | BRIDGE | Diffusion-TS | TimeGAN | TimeVAE | TimeVQVAE | GT-GAN |
> | --- | --- | --- | --- | --- | --- | --- |
> | Inference | 27.7 | 142.8 | 6.3 | 3.84 | 4.75 | 7.23 |
> | Training | 356.6 | 654.7 | 723.1 | 16.67 | 157.1 | 454.3 |
>
> **[Q 3] Can BRIDGE generalize to unseen domains without fine-tuning? Are there cases where prototype alignment fails?**
>
> Our framework was explicitly designed to promote cross-domain generalization by leveraging semantic prototypes and text conditioning. In our experiments (see Table 3), **BRIDGE was able to generate high-fidelity time series in unseen domains (e.g., Stock, Web) without any fine-tuning**, suggesting that the learned semantic structure and prototype conditioning do generalize beyond the training domains.
>
> While we did not observe clear prototype misalignment in these settings, we agree that such cases may be possible, particularly in domains with very different temporal dynamics or semantics. We will explore this further in future work by evaluating on a broader set of domains and investigating potential prototype refinement strategies.
>
> ---
>
> Thank you again for your thoughtful advice. We will certainly incorporate the discussions in the revised version and include references on prompt optimization and LLM-based dataset generation as per your suggestion.
> Please feel free to raise any further questions—we greatly appreciate your ongoing input!

---

> > ### Comment · Reviewer_svQz · 2025-04-08
> >
> > Thanks to the author for the reply, which have addressed my concerns. I have raised my score.

---

> > > ### Author Response · Authors · 2025-04-09
> > >
> > > Thank you so much for your kind and supportive feedback. We truly appreciate the time you took to review our submission and response. Your encouraging comments and support in raising the score mean a great deal to us.

---

### Official Review · Reviewer_QJJi · 2025-03-19

**Overall Recommendation:** 3

**Summary:**

The paper considers how to train a generator for time series (the time series generation problem). A multi-step system (process) is created, that includes LLM prompts, time series features and specification, and training.  There is also a two-team multi-agent method embedded for some optimization within the system.  Several time series data are used, and numerical comparisons are made to some other methods. The primary contribution is some generalization across different time series training data. The numerical results are somewhat mixed in various measures of success.

## update after rebuttal

The authors have provided useful explantation and discussion of their work.  The overall concept and approach is interesting, and I have raised my score.  However, I am still skeptical that the time series application is a good one for this work, and believe that a good generator can be trained without using the semantic approach.

**Claims And Evidence:**

Claims of generality are not very clear.  The time series specifications (or features) are hand crafted and a rich enough set of these is selected so as to be sufficient for modeling the various time series data.


There is little or no formal theory or convergence analysis, the results are entirely based on tuning the system to work with the data sets considered.  The extend and impact of the human interventions isn't clear, and this leaves the reader wondering about the overall approach and outcomes.

**Essential References Not Discussed:**

The paper is not well linked with (decades of) time series literature.

**Experimental Designs Or Analyses:**

It is not clear how to evaluate the many aspects of filtering, human interactions, and selected time series features.

**Methods And Evaluation Criteria:**

The data sets are varied to some extent.  However, it isn't clear that existing time series signal decomposition and feature extraction methods couldn't be applied to this data, and use this for training a generator.   The testing here is against other similar algorithms, and the advance seems limited to being somewhat more generalized over the example set.


The system is so highly tuned with so many pieces that, although the entire approach seems reasonable, the reader can't easily judge the overall contribution or outcomes.


Overall, it simply isn't clear that a semantic interface is a good idea for this problem.  Having said that, the paper is most interesting in terms of the overall system approach, but the contribution to the application (time series analysis and generation) is very small.

**Other Comments Or Suggestions:**

Would be very good to clearly show the hand tuned and human interaction portions.

Can this framework be applied to other applications?

Why is the multi-agent approach a good one?  Isn't this just an optimization problem?  What other optimizers could be applied?

**Other Strengths And Weaknesses:**

Please see comments in other sections.

***Added after rebuttal.***

The authors have provided useful explantation and discussion of their work.  The overall concept and approach is interesting, and I have raised my score.  However, I am still skeptical that the time series application is a good one for this work, and believe that a good generator can be trained without using the semantic approach.

********************************

**Questions For Authors:**

Given the hand crafted selection of time series specifications, then why is the LLM-based approach a good one?

**Relation To Broader Scientific Literature:**

The contribution to time series analysis and generation is minor.  This is because the features and specifications needed amount to many constraints and properties of the observed time series, and these are hand crafted until the semantic specification of these for an LLM leads to something useful.  So if the point is to show an example of using an LLM with a generator, then the approach is somewhat interesting.  But if the true goal is a time series generator, then adding in an LLM creates more ambiguity than it solves, and alternative methods for training generators should be found.

**Theoretical Claims:**

Not applicable.

---

> ### Author Rebuttal · Authors · 2025-03-31
>
> We appreciate the reviewer's time and feedback, especially the high-level perspective on the entire system, which will guide the future expansion of our work. Thank you!
>
> ####  **General Comments (GC):**
>
> **[GC 1]** *Claims of generality are not very clear.* *"The time series specifications (or features) are hand-crafted, and a rich enough set of these is selected."*
>
> Section 4.1 generates text templates, which are then applied to TS data, with **an LLM filling in general, domain-agnostic statistical features** to produce the final text. Thus, **time series specifications are automatically generated by the LLM based on templates** from the multi-agent system, **rather than being manually crafted or selected by human intervention**. We will make it more clear in future revisions.
>
> **[GC 2]** *"The extent and impact of human interventions are unclear."*
>
> There are only two points in our work where human actions are involved:
>
> 1. After collecting the initial text templates **(Section 4.1 Step1)**, LLM prompting filters them to ensure no dataset-specific information is included. A human double-checks the results to prevent potential data leakage.
> 2. A human assessment (Section 6.2) serves only as an evaluation metric and does not affect model training.
>
> **[GC 3]** *"It isn't clear that existing time series signal decomposition and feature extraction methods couldn't be applied to this data and used for training a generator."*
>
> We tried rule-based text generation and Seasonal-Trend Decomposition, but both methods were ineffective in generating useful descriptions (**the first paragraph in Section 4 and Appendix A9**).
>
> **[GC 4]** *"The contribution to time series analysis and generation is very small."*
>
> Our framework is designed specifically to addresses key challenges in text-controlled TS generation, such as lack of aligned data, controllable generation, and domain generalization. We integrate multi-agent reasoning for data construction, semantic prototype conditioning, and a diffusion-based generator to address challenges.
>
> Within the multi agent system, we propose tailored evaluation criteria like trend description accuracy, seasonality, completeness, and clarity (Appendix A.5) to assess TS generation. We also explore the impact of different text types on generation, showing that pattern descriptions and concise summaries are more effective (Section 6.3). These insights offer valuable takeaways for time series tasks.
>
> ####  **Other Comments (OC):**
>
> **[OC 1]** *"It would be very helpful to clearly show the hand-tuned and human interaction portions."*
>
> We clarify that there is no hand-tuning or human intervention for time series features.
>
> Please refer to our response to **[GC 1]** and **[GC 2]**.
>
> **[OC 2]** *"Can this framework be applied to other applications?"*
>
> Great suggestion! The modular structure could support future applications beyond TS Generation, including in image and video. This is an exciting direction for future work.
>
> **[OC 3]** *“Why is the multi-agent approach a good one? Isn't this just an optimization problem? What other optimizers could be applied?”*
>
> We chose a multi-agent system over a single-agent or LLM prompt approach for key reasons. Direct LLM prompting was ineffective in generating useful descriptions (see **[GC 3]**), and single-agent systems often suffer from overconfidence, leading to errors without correction [1]. Multi-agent systems allow agents to critique each other’s outputs, improving reliability and transparency by assigning distinct roles.
>
> Our multi-agent approach goes beyond scalar optimization, involving tasks like searching for TS text online, summarizing templates, generating text, evaluating alignment, and refining for better templates.
>
> While  we recognize reinforcement learning could optimize a single LLM, it faces challenges: (1) large optimization space, as RL starts from scratch, while our method refines existing templates; (2) high computational demands due to model training, whereas our agents require no retraining or fine-tuning; and (3) sparse feedback from using final generation performance as a reward, while our agents themselves evaluate text clarity, completeness etc (Appendix A5), improving refinement efficiency.
>
> [1] Enhancing LLM Reasoning with Multi-Path Collaborative Reactive and Reflection agents [[arxiv.org/pdf/2501.00430](https://arxiv.org/pdf/2501.00430)]
>
> #### **Question**:
>
> *“Given the hand crafted selection of time series specifications, then why is the LLM-based approach a good one?”*
>
> Please kindly refer to our response to **[GC 1], [GC 2]** for the clarification on avoiding human intervention in feature selection and **[OC 3]** for justification of the multi-agent system. As general statistics with simple rule-based texts are ineffective for TSG (see Section 6.3), we use a multi-agent system.
>
> If any further questions or concerns, we’d be happy to provide more information. Looking forward to your response and continued discussions!

---

> > ### Comment · Reviewer_QJJi · 2025-04-08
> >
> > Thank you for your response to my and the other reviews.  These detailed comments help with understanding. Based on the reviews and response, I will raise my score.  I think that the overall ideas and approach represents interesting research, and although I'm not convinced that time series is a useful problem to study, nevertheless the general research approach is interesting.
> >
> > Some comments about the time series aspects:
> >
> > I do continue to have concerns about the general motivation for this particular application, and feel that the major contribution and interesting research is more about the overall method and approach, and not the focus on time series.
> >
> > I am skeptical that creating good generators for these various time series requires any kind of text based reasoning.  It isn't clear that the 'rule based' comparison is very useful.
> >
> > The method relies on internet-available information with semantic descriptions of time series aspects that must be related back to mathematics.  So there can be no real guarantees of reliability, and perhaps even bad influences could come into the picture. If a time series generation problem were of significant interest, then domain expertise would be needed to bring such guarantees.
> >
> > It isn't clear that the work brings any new 'features' or TS characteristics that aren't already explored in TS literature.
> >
> > The meta-agent framework is interesting, although it seems other recent frameworks could also be used.  Consequently, we have an ML research problem within an ML research problem.  This is a challenge for both the authors and the reviewers.

---

> > > ### Author Response · Authors · 2025-04-09
> > >
> > > We sincerely thank the reviewer for taking the time to engage with our work in such depth, offering constructive comments, and kindly raising the score. We truly appreciate your thoughtful suggestions, which have been valuable in helping us improve both the clarity of our presentation and the direction of our ongoing efforts.
> > >
> > > Your question regarding the general motivation of our work and your suggestion that the framework may extend beyond time series are both insightful and encouraging. Controllable time series generation is itself an important and challenging research problem, with broad real-world relevance. We aim at using text to make the generation process more interpretable and generalizable, which are important in supporting downstream tasks. While we focused on time series due to the unique challenges of controllable generation in this domain, we fully agree that the underlying methodology could extend to other modalities like image and video. This is a direction we find exciting and will actively explore in future iterations of this work.
> > >
> > > We also welcomed your thoughtful questions on the role of domain expertise in using external textual sources and the novelty of the TS features. Strengthening reliability in sensitive domains such as clinical applications is an important future direction, and we will work on extending our framework to incorporate domain expertise. While we do not explicitly introduce new TS features, our use of text embeddings effectively enriches TS representation and supports controllable generation. We believe that further exploring how textual abstractions interact with traditional TS features is a promising direction for future work and one that could offer meaningful contributions to the community.
> > >
> > > Thank you as well for your comments on the multi-agent design. We will include the experimental results and discussions mentioned in our response to Reviewer cEkx to further clarify this component. Additionally, we plan to explore simpler or more targeted variants that retain interpretability while reducing system complexity.
> > >
> > > Once again, thank you for your generous feedback and thoughtful suggestions. Your comments have helped us sharpen the scope of our work, identify promising extensions, and consider broader implications. We are very grateful for your support in improving this paper.

---

### Decision · Program_Chairs · 2025-05-01

**Decision:**

Accept (poster)

**Comment:**

This paper introduces a framework for text-controlled time-series generation, tackling the important challenges of controllability and data scarcity in this domain. One of the main contributions in the paper is the LLM-based multi-agent system designed to bootstrap high-quality text-to-time-series paired datasets. The authors present interesting empirical results across various datasets, demonstrating state-of-the-art generation on most datasets and significant improvements in controllability compared to unconditional generation or generation without text. Reviewers generally said the task formulation was novel and that the combination of multi-agent systems and diffusion models is interesting. Initial reviews raised some concerns about the evaluation metrics (e.g., a lack of standard generative metrics), as we as comparison to recent related work (e.g., Diffusion-TS, Time-MMD).
The authors provided a thorough rebuttal, addressing most of their concerns. Particularly, they added extensive experiments, new metrics (PS, DS, J-FTSD), comparisons to requested baselines/frameworks, and detailed clarifications that successfully addressed most major concerns raised by the reviewers.  While minor points regarding the specific implementation choices still remain, the overall methodology is sound, technically solid, and well-executed. Overall, this paper represents a solid contribution to controllable time-series generation.